# Epigenetic reprogramming at estrogen-receptor binding sites alters 3D chromatin landscape in endocrine-resistant breast cancer

Joanna Achinger-Kawecka [1,2], Fatima Valdes-Mora [1,2], Phuc-Loi Luu[1,2], Katherine A. Giles[1], C. Elizabeth Caldon[3], Wenjia Qu[1], Shalima Nair[1], Sebastian Soto[1], Warwick J. Locke[1], Nicole S. Yeo-Teh[1], Cathryn M. Gould[1], Qian Du[1], Grady C. Smith[1], Irene R. Ramos[4], Kristine F. Fernandez[3], Dave S. Hoon [4], Julia M.W. Gee[5], Clare Stirzaker[1,2] & Susan J. Clark [1,2]*

Endocrine therapy resistance frequently develops in estrogen receptor positive (ER+) breast cancer, but the underlying molecular mechanisms are largely unknown. Here, we show that 3-dimensional (3D) chromatin interactions both within and between topologically associating domains (TADs) frequently change in ER+ endocrine-resistant breast cancer cells and that the differential interactions are enriched for resistance-associated genetic variants at CTCF-bound anchors. Ectopic chromatin interactions are preferentially enriched at active enhancers and promoters and ER binding sites, and are associated with altered expression of ER-regulated genes, consistent with dynamic remodelling of ER pathways accompanying the development of endocrine resistance. We observe that loss of 3D chromatin interactions often occurs coincidently with hypermethylation and loss of ER binding. Alterations in active A and inactive B chromosomal compartments are also associated with decreased ER binding and atypical interactions and gene expression. Together, our results suggest that 3D epi-genome remodelling is a key mechanism underlying endocrine resistance in ER+ breast cancer.

[1] Epigenetics Research Laboratory, Genomics and Epigenetics Theme, Garvan Institute of Medical Research, Sydney, NSW 2010, Australia. [2] St. Vincent's Clinical School, Faculty of Medicine, UNSW Sydney, Sydney, NSW 2010, Australia. [3] Cancer Theme, The Kinghorn Cancer Centre, Sydney, NSW 2010, Australia. [4] Department of Translational Molecular Medicine, John Wayne Cancer Institute, Santa Monica, CA, USA. [5] Breast Cancer Molecular Pharmacology Group, School of Pharmacy and Pharmaceutical Sciences, Cardiff University, Wales CF10 3NB, UK. *email: s.clark@garvan.org.au

nappropriate reprogramming of the estrogen receptor (ER) signalling network in mammary epithelial cells initiates neoplastic transformation and drives ER-positive (ER+) breast cancer[1]. ER+ breast cancer patients will receive long-term endocrine therapies, including tamoxifen and fulvestrant, to inhibit the ER-signalling pathways on which their tumours are dependent[2]. Tamoxifen is one of the selective estrogen receptor modulators (SERM) and acts as an antiestrogen in the mammary tissue whereas fulvestrant is a selective estrogen receptor degrader (SERD) and acts by binding to the estrogen receptor and destabilising it[3]. Endocrine treatment significantly reduces the relapse rate by almost 50%; however, around 33% of patients treated with endocrine therapy develop endocrine resistance and relapse within 15 years of first receiving treatment[4]. The molecular factors that define endocrine response in ER+ breast cancer patients remain poorly understood. Recent research suggests that global reprogramming of estrogen-responsive regions of the genome can modulate endocrine sensitivity and contribute to the onset of ER+ breast cancer and the acquisition of endocrine resistance[5–7]. We also reported that differential DNA methylation at estrogen-responsive enhancers is associated with endocrine response in breast cancer[6], raising the possibility that dynamic three-dimensional (3D) chromatin remodelling of ER-mediated enhancer−promoter interactions could potentially underlie the development of endocrine resistance in breast cancer. However, these and related studies[6,8] did not directly interrogate the role of 3D genome architecture in association with such alterations.

Chromosome conformation capture (Hi-C) interaction maps provide information on multiple levels of 3D chromatin structure and three main layers of genome organisation have been described to date[9,10]. First, at the level of single genes, the genome is organised into local enhancer–promoter interactions. Second, the genome is segmented into topologically associated domains (TADs) that are ~1 Mb in size and encompass multiple genes and regulatory elements[10,11]. TADs are conserved and largely invariant between different cell types, while the chromatin interactions within TADs are more tissue specific[12–14]. Finally, at the higher level, chromatin interactions and TADs are organised into functionally distinct compartments, comprising large regions that are either A-type (active) or B-type (inactive)[9].

Here, we characterise the 3D chromatin organisation in endocrine-sensitive and endocrine-resistant ER+ breast cancer cell lines. We show that 3D epigenome remodelling is a key mechanism associated with endocrine resistance that consists of aberrant DNA methylation and differential ER-bound enhancer −promoter interactions.

## Results

**Differential interactions associate with altered expression.** To initially address if 3D epigenetic remodelling is associated with endocrine resistance, independent of the class of endocrine therapy, we performed in situ Hi-C experiments in parental endocrine-sensitive ER+ MCF7 cells, tamoxifen-resistant (TAMR)[15] cells and fulvestrant-resistant (FASR)[16] cells (Supplementary Table 1). Multidimensional scaling analysis of Hi-C contact maps revealed high dissimilarity between MCF7, TAMR and FASR genomes (Fig. 1a). As a control to ensure that the differences in the chromatin contacts were associated with endocrine resistance, and not due to long-term culture, we performed Hi-C on MCF7 cells, grown without exposure to endocrine therapy, at three time points; start (T0), culture mid-point (3 months: T3) and late culture (6 months; T6). MDS analyses of Hi-C contact maps at multiple resolutions (5 Mb, 1 Mb and 100Kb) show that all the MCF7 cells, including the public Hi-C

MCF7 data[17] cluster away from the resistant derivatives TAMR and FASR (Supplementary Fig. 1a), irrespective of the time in culture.

Next to identify the differential chromatin interactions between the parental MCF7 cells and endocrine-resistant cells we used the diffHiC method[18], and found 981 significantly different interactions between MCF7 and tamoxifen-resistant TAMR cells (diffHiC, FDR < 0.05, Supplementary Data 1) and 2596 significantly differential interactions between MCF7 and fulvestrant-resistant FASR cells (diffHiC, FDR < 0.05, Supplementary Data 2) at 20 kb resolution. Differential interactions were more often lost with the development of fulvestrant resistance (62% are MCF7-specific), while there were similar numbers of differential interactions lost and gained in the tamoxifen-resistant cells (46% are TAMR-specific) (Fig. 1b). The majority of differential interactions detected in TAMR cells were not present in FASR cells (Fig. 1b), potentially consistent with the different mode of action between tamoxifen and fulvestrant and the different pathways to development of endocrine resistance in these two models[15,16].

Since 3D chromatin interactions bring distal regulatory elements, such as enhancers into close proximity of their target genes, we explored whether differential interactions gained and lost in endocrine resistance include direct enhancer−promoter interactions. We integrated the differential chromatin interaction data with chromatin state information based on five ChIP-seq marks (H3K27ac, H3K4me1, H3K4me3, H2AZac and H3K27me3) using chromHMM[19]. Interestingly, all differential interactions were significantly enriched for enhancer and promoters, as well as CTCF sites, regardless of the TAMR or FASR treatment regime (Fig. 1c). However, gained chromatin interactions in TAMR and FASR cells showed higher enrichment of active enhancer marks (H3K4me1 and H3K27ac), compared to lost interactions (Supplementary Fig. 1b). Similarly there was increased enrichment of the active promoter mark H3K4me3 at gained interactions in TAMR and FASR cells relative to MCF7 cells (Supplementary Fig. 1b).

Differential interactions are frequently associated with altered expression of the genes they connect[20,21]. Therefore, to address whether differential interactions present in endocrine-resistant cells are associated with deregulation of gene expression, we identified genes located at anchors of differential interactions and compared their expression between MCF7 cells and TAMR and FASR cells (Supplementary Data 3). We found that differentially expressed genes were enriched for pathways known to be associated with endocrine resistance (e.g. estrogen response) and cancer (e.g. EMT, angiogenesis) (Supplementary Fig. 1c). Differential interactions in FASR cells overlapped promoters of 2069 genes and loss of interactions in these cells was associated with decreased expression of 213 genes (Fig. 1d). Gain of interactions was associated with increased expression of 170 genes (Fig. 1e). In TAMR cells, 500 genes were located at differential chromatin interactions. Loss of interactions resulted in significant decrease in gene expression, with 50 genes downregulated (Supplementary Fig. 1d), while gained interactions were associated with increased expression of 21 genes (Supplementary Fig. 1e). Overall, in both TAMR and FASR cells, lost and gained interactions were enriched for differentially expressed genes (FDR < 0.05) (chi-square test P < 0.001) with most of the genes located at anchors of lost interactions being downregulated, and genes located at ectopic/gained interactions being upregulated. Interestingly, genes present at differential chromatin interactions in TAMR cells were often enriched for similar Gene Ontology terms as genes present at differential interactions in FASR cells. This included transcription, cell−cell adhesion and G2/M transition (Supplementary Fig. 1f). Additionally, some of

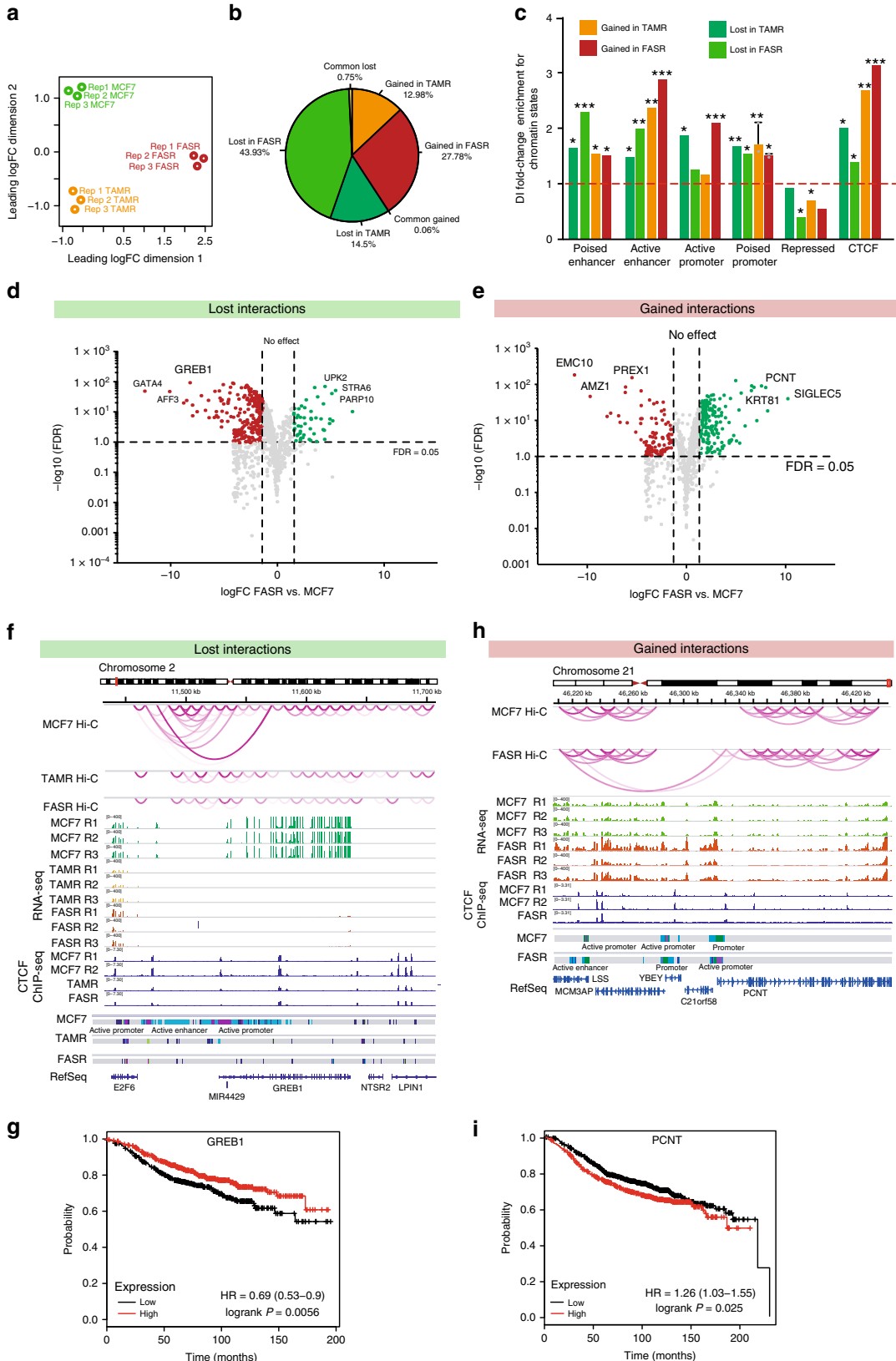

the enriched GO terms were specific to either TAMR or FASR. Specifically, genes at TAMR differential interactions were related to Erbb2 signalling pathways, response to estradiol and Wnt receptor signalling, and FASR-specific differential interactions were enriched for terms related to apoptosis, MAPK cascade, cell division and migration (Supplementary Fig. 1f).

Figure 1f shows a representative example of differential interaction that is lost in both TAMR and FASR cells as compared to MCF7 cells. In MCF7 cells, the active promoter of the *GREB1* gene is connected via strong long-range interactions with a putative distal active enhancer and the *GREB1* gene is strongly expressed. In TAMRs and FASRs these distal enhancers

**Fig. 1 Differential enhancer−promoter interactions and gene deregulation. a** Multidimensional scaling plot (MDS) of the top 1000 interactions for each individual Hi-C replicate (MCF7, TAMR and FASR) at 20 kb resolution. **b** Pie chart showing overlapping anchor regions between differential interactions identified in TAMR vs. MCF7 diffHiC and FASR vs. MCF7 diffHiC analysis. **c** Differential interactions (DIs) enrichment for chromatin states based on ChromHMM segmentation and transcription factor binding at lost and gained differential interactions in TAMR and FASR cells as compared to MCF7 cells. Asterisks represent the significance of fold-change enrichment at observed vs. random regions (permutation test, SD, $n = 2$). ***$P$ value < 0.001, **$P$ value < 0.005, *$P$ value < 0.05. **d** Volcano plot (−log10FDR vs. log2 fold change) of all genes present at anchors of lost differential interactions between FASR and MCF7 cells. Source data are provided as a Source Data file. **e** Volcano plot (−log10FDR vs. log2 fold change) of all genes present at anchors of gained differential interactions between FASR and MCF7 cells. Source data are provided as a Source Data file. **f** Representative example demonstrating the association between enhancer−promoter interactions lost in TAMR and FASR cells as compared to MCF7 cells and decreased expression of *GREB1* gene. Numerous interactions between active enhancers and active promoter of *GREB1* gene are present in MCF7 cells. In TAMR and FASR cells this region is occupied by poised enhancers and the long-range interaction present in MCF7 cells is lost. CTCF ChIP-seq track is shown. **g** Kaplan−Meier curves displaying relapse-free survival for 742 patients with ER+ tumours receiving endocrine treatment based on *GREB1* gene expression. Patients with tumours with high expression of *GREB1* are shown in red and those with low expression are shown in black. *P* value as indicated, log rank test. **h** Representative example demonstrating the association between enhancer−promoter interactions gained in FASR cells as compared to MCF7 cells and overexpression of *PCNT* gene. Long-range interactions between distant enhancer and promoter of *PCNT* gene are present in FASR cells and absent in MCF7 cells. CTCF ChIP-seq track is shown. **i** Kaplan−Meier curves displaying relapse-free survival for 742 patients with ER+ tumours receiving endocrine treatment based on *PCNT* gene expression. Patients with tumours with high expression of *PCNT* are shown in red and those with low expression are shown in black. *P* value as indicated, log rank test.

become poised or are lost and only local interactions are present at the *GREB1* gene locus, and the *GREB1* gene is inactive (Fig. 1f and Supplementary Fig. 2a). Notably, expression of *GREB1* is a predictive biomarker for breast cancer endocrine resistance[22,23] and low expression of *GREB1* is associated with reduced relapse-free survival in 742 patients with ER+ breast cancer treated with endocrine therapy[24] (logrank $P = 0.0056$) (Fig. 1g). Figure 1h shows an example of a *PCNT* gene locus, where multiple local interactions between the *PCNT* promoter and intragenic enhancers are gained in FASR cells as compared to MCF7 cells. This is associated with an increase in *PCNT* expression in FASR (Fig. 1h and Supplementary Fig. 2b). High expression of *PCNT* is also significantly associated with reduced relapse-free survival in ER+ breast cancer patients treated with endocrine therapy[24] (logrank $P = 0.025$) (Fig. 1i). Further examples demonstrating the association between differential interactions, change in chromatin states and gene expression in FASRs are presented in Supplementary Fig. 2c, d. Examples of enhancer−promoter interactions in TAMR cells that are lost (Supplementary Fig. 2e, f) or gained (Supplementary Fig. 2g, h), and are associated with change in chromatin and deregulation of gene expression are also shown. Together the data show that new enhancer and promoter interactions occur in both tamoxifen- and fulvestrant-resistant cells that are associated with differential expression of genes they connect.

**SNVs at CTCF sites associate with loss of interactions.** By integrating single-nucleotide variation (SNV) data obtained from whole genome sequencing (WGS) with differential interactions identified in TAMR and FASR cells, we investigated whether SNVs uniquely present in endocrine-resistant cells could directly alter individual differential 3D interactions. Using MuTect2, we defined putative endocrine resistance-associated SNVs as variants present only in TAMR/FASR cells and absent in parental MCF7 cells (see Methods and Supplementary Table 2). We then gathered the coordinates of SNVs associated with endocrine resistance (14,652 in TAMR and 15,381 in FASR) and annotated their location in relation to genes. Both in TAMR and FASR cells, ~60% of resistance-associated SNVs mapped to intergenic regions, ~35% to introns and ~1% to coding exons (182 in TAMR and 139 in FASR) (Supplementary Fig. 3a). By estimating the somatic mutation rate of SNVs in each 100 kb bin of the genome, we found that anchors of gained differential interactions in TAMR and FASR cells have elevated levels of SNVs as compared to matched, randomised regions (see Methods and ref. [25])

(Supplementary Fig. 3b). Anchors of differential interactions that were lost in TAMR and FASR cells were depleted for SNVs as compared to random regions (Supplementary Fig. 3b). Similarly, we observed that SNVs were highly enriched at differential interactions that were lost in TAMR and FASR cells, but depleted at gained interactions (Fig. 2a), suggesting that SNVs are involved in loss of differential chromatin interactions in ER+ endocrine-resistant cells. To establish the putative mechanism of SNVs role in differential interactions, we first investigated the distribution of endocrine-resistance-associated SNVs across the epigenome. We determined that the SNVs in TAMR cells are enriched at DNase hypersensitive sites (DHS) (ENCODE), while FASR SNVs did not show genome-wide enrichment (Supplementary Fig. 3c). To capture SNVs in regulatory elements, we assessed the enrichment of SNVs at a total of 85 transcription factors binding sites using ReMap database[26]. We observed that resistance-associated SNVs were strongly enriched at HSF1, NCOA1/2/3, ESR1, CTCF and FOXA1 binding sites in both TAMR and FASR cells (Fig. 2b). Additionally, SNVs in TAMR cells were highly enriched at AR and RAD21 biding, while SNVs in FASR cells were enriched at GATA3 (Fig. 2b).

Previous studies have shown that genetic mutations at CTCF binding sites can alter the long-range chromatin loops and TAD boundaries anchored by these sites[27−30]. Therefore, we next overlapped resistance-specific SNVs present at anchors of differential interactions with CTCF ChIP-seq peaks lost in TAMRs or FASRs and identified 28 resistance-associated SNVs located at lost CTCF binding sites and associated with differential interactions in TAMRs and 46 resistance-associated SNVs in FASRs. In both TAMR and FASR cells, more resistance-associated SNVs were located at anchors of interactions that were lost in resistant cells, as compared to interactions that were gained (Fig. 2c). All of identified SNVs were located within the CTCF ChIP-seq peak that is lost, but only three (one in TAMRs and two in FASRs) were directly located within the CTCF motif (Fig. 2d), suggesting that genetic perturbation close to CTCF sites also leads to a decrease in interactions. Figure 2e shows an example of putative resistance-associated SNVs on chromosome 17, located at the anchors of differential chromatin interactions present in MCF7 cells that are lost in FASR cells (Fig. 2e). One of these SNVs (rs201722399) is located within the CTCF motif that is strongly bound by CTCF in MCF7 cells, but loses its binding in FASR cells (highlighted in orange and in the zoomed-in view). Additionally, three other resistance-associated SNVs are present that are located at CTCF binding sites (highlighted in yellow).

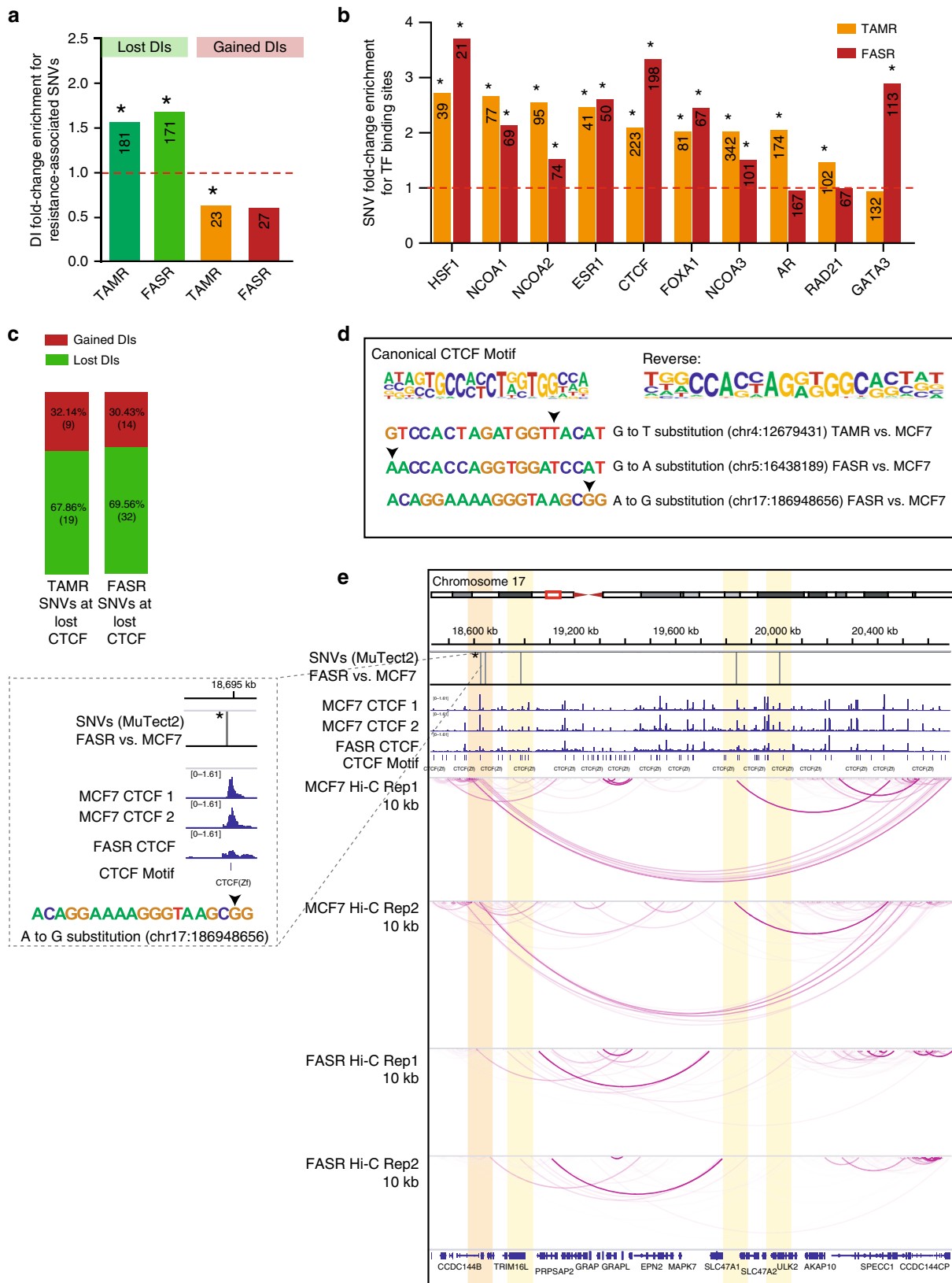

Further examples of SNVs located at anchors of differential interactions on chromosome 4 and 5 that are within CTCF binding sites in TAMR and FASR cells are shown in Supplementary Fig. 3d, e. Together these data support a direct link between genetic alterations at CTCF motifs and loss of CTCF binding and loss of interactions.

To demonstrate the potential role of endocrine-associated SNVs at differential interactions on gene regulation, we defined candidate target genes based on detected interactions between SNVs and gene promoters. This resulted in 33 candidate target genes at 74 resistance-associated SNVs located at lost CTCF binding sites and associated with differential interactions in both

**Fig. 2 SNVs associate with loss of CTCF binding and loss of interactions. a** Differential interactions (DIs) enrichment for SNVs identified by MuTect2 in TAMR and FASR genomes as compared to MCF7 genome. Asterisks represent the significance of fold-change enrichment at observed vs. random regions (permutation test). *P value < 0.005. The numbers of DIs located within each specific region are presented in the respective column. **b** Endocrine-resistance-associated SNVs fold-change enrichment for transcription factor binding sites from ReMap v.2 2018. The numbers of SNVs located within each specific binding site are presented in the respective column (*P value < 0.005). **c** Proportion of resistance-associated SNVs associated with loss of CTCF binding at anchors of gained or lost differential interactions in endocrine-resistant cell lines as compared to MCF7 cells. **d** Three different resistance-associated SNVs affecting the CTCF binding motif at anchors of differential interactions in TAMR and FASR cells. Arrow points to a nucleotide substitution in the CTCF motif obtained from Homer. **e** MCF7 (top) and FASR (bottom) Hi-C interactions map at 10 kb resolution on chromosome 17 showing a differential interaction, which is lost in FASR cells and associated with an SNV (rs201722399) located within a CTCF binding motif (marked with black arrow) that is lost in FASR cells (highlighted in orange and in the zoomed-in view). Additional three resistance-associated SNVs are located within this region that overlap a CTCF binding site (highlighted in yellow).

TAMRs and FASRs, including 29 protein-coding genes (Supplementary Data 4). Importantly, out of these, 9 genes were differentially expressed in TAMR cells and 15 were differentially expressed in FASR cells (Supplementary Fig. 3f) (t test with Benjamini−Hochberg FDR). Most candidate target genes interacted with only one SNV, but three genes interacted with more than two SNVs, including ZNF622 on chromosome 5, which interacts with three SNVs in FASR cells (Supplementary Fig. 3e). Our analyses provides a list of putative target genes for future functional validations and illustrates the power of using Hi-C to link genetic variants to potential target genes.

**Differential interactions occur at altered ER binding sites**. We next asked if anchors of differential interactions are enriched for binding motifs of other known transcription factors by using in silico motif analysis with Homer (see Methods). We found that in tamoxifen resistance, interactions lost in TAMR cells mainly occurred at regions of ERE, SOX2, FOXA1 and HOX cluster (HOXB13 and HOXD13) binding motifs, while interactions gained in TAMR were present at ZNF143, OCT4, FOXA1 and RUNX2 binding motifs (Fig. 3a). This is in agreement with the current model in which tamoxifen resistance results in aberrant ER signalling throughout the genome[31]. Additionally, ZNF143 has been recently suggested as a cofactor in regulation of chromatin looping[32]. In fulvestrant resistance, interactions lost in FASR were mainly present at SOX6, NRF2 and ATF3 binding motifs, while gained interactions were enriched for OTX2 and SMAD4 binding motifs (Supplementary Fig. 4a). Interestingly, differential interactions in both models of endocrine resistance occur commonly at c-Myc binding motifs (Fig. 3a and Supplementary Fig. 4a), confirming an important role for c-Myc activation in the overall development of endocrine resistance, as previously suggested in early breast carcinogenesis model[33,34] and patient outcome studies[35]. To further validate the in silico predictions, we assessed the enrichment of transcription factor (TF) binding in MCF7 cells[26] at anchors of differential interactions. We observed large number of significantly enriched TF at differential interactions (P value < 0.001 and FC > 2, permutation test) (Supplementary Data 5). ESR1, MYC, CTCF, NR2F1, FOXA1 and PgR were most commonly observed. Out of 22 transcription factor motifs identified in in silico analyses, public ChIP-seq data were available for seven TF. Out of these six (i.e. MYC, NRF2, ARTN2, FOXA1, ESR1, PgR) showed significant observed/expected enrichment for binding at anchors of differential interactions, validating our in silico predictions. Interestingly, ZNF143 binding was not enriched at anchors of differential interactions, despite strong prediction for its motif at both gained and lost interactions in TAMRs. This is consistent with its role as a cofactor instead of pioneer factor at chromatin loops[32].

Estrogen receptor is the defining and driving transcription factor in the majority of breast cancers and its target genes dictate endocrine sensitivity status of ER+ breast cancer cells[36]. Long-

term culture of MCF7 cells with fulvestrant is associated with a complete suppression of estrogen receptor signalling[31], which is maintained after development of resistance (ESR1 logFC = −4.31, FDR < 0.001, Fig. 3b), suggesting that it is not directly participating in the development and maintenance of fulvestrant resistance. However, in tamoxifen resistance, ESR1 expression is decreased but still present (logFC = −2.09, FDR < 0.001) (Fig. 3b) after the cells become resistant, consistent with previous work that showed many changes in ER-regulated signalling and ER binding in TAMR cells[5,7,37]. To test whether remodelling of ER binding is also associated with the formation of differential interactions that are altered in endocrine resistance, we utilised ER ChIP-seq data from MCF7 and TAMR cells[37]. We compared ER binding sites between MCF7 and TAMR cells and determined a subset of ER binding sites that were gained in TAMR, lost in TAMR or common in both MCF7 and TAMR (see Methods). We identified 14,002 common ER binding sites between the two cell types, with 15,447 ER binding sites unique to MCF7 cells (i.e. lost in TAMRs) and 5395 ER binding sites unique to TAMR cells (i.e. gained in TAMRs) (Fig. 3c). Differential interactions gained in TAMR were highly enriched for gained ER binding, and depleted for lost ER binding sites and interactions lost in TAMR were enriched for lost ER binding events (Fig. 3d and Supplementary Fig. 4b). Both lost and gained differential interactions showed a significant enrichment at common ER binding sites between MCF7 and TAMR cells (Fig. 3d). We next examined the proportion of differential interactions that are present at altered and common ER binding sites and observed that 234 of interactions gained in TAMR were located at gained ER binding sites and 307 of interactions lost in TAMR were located at lost ER binding sites (Fig. 3e). Interestingly, ER-bound differential interactions gained in TAMR cells were strongly enriched for several Gene Ontology terms relating to epidermal growth factor receptor signalling pathway (Fig. 3f). Epidermal growth factor receptor/ErbB2 gene expression is significantly increased in TAMR cells as compared to MCF7 (logFC = 5.6, P value < 0.0001). This is in agreement with previous studies showing ER-mediated repression of ErbB2 is a hallmark of tamoxifen sensitivity in breast cancer[38].

**Methylation associates with ER loss and enhancer interactions**. Previously we have shown that DNA hypermethylation in endocrine-resistant cells is associated with loss of ER binding at enhancer regions[6]. To confirm this result genome-wide, we characterised the DNA methylation patterns in MCF7, TAMR and FASR cells using whole genome bisulphite sequencing (WGBS). As shown previously[6], we observed marked DNA hypermethylation in TAMR cells as compared to parental MCF7 cells with 90% of differentially methylated regions (DMRs) showing an increase in methylation, whereas in FASR cells only 20% of the DMRs were hypermethylated. We next asked whether these endocrine-resistance-specific changes in DNA methylation are associated with differential ER binding and differential

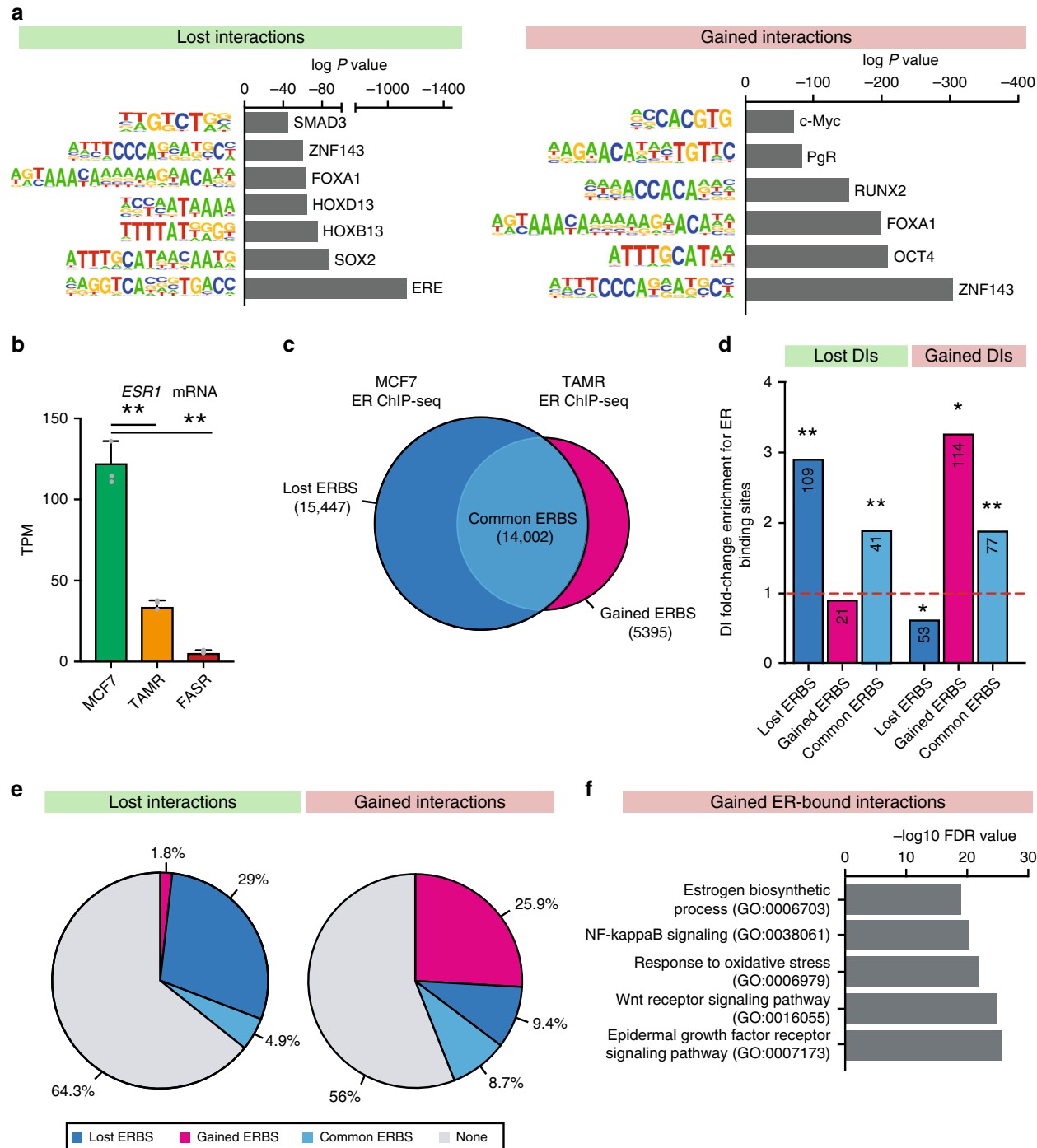

**Fig. 3 Differential interactions occur at altered ER binding sites. a** Motifs enriched at anchors of lost (left panel) and gained (right panel) interactions between TAMR and MCF7 cells. Known motifs obtained from Homer database and compared to matched, randomised background regions. **b** *ESR1* mRNA expression downregulated in TAMR cells and lost in FASR cells (**P value < 0.001, SD, n = 3). **c** Venn diagram showing lost and gained ER binding sites (ERBS) in TAMR cells as compared to parental MCF7 cells identified with diffBind. ChIP-seq data obtained from Ross-Innes et al.[37]. **d** Differential interactions (DIs) enrichment for ER binding sites lost and gained between TAMR and MCF7 cells. Asterisks represent the significance of fold-change enrichment at observed vs. random regions (permutation test). **P value < 0.001, *P value < 0.005. The numbers of ER binding sites located within each specific region are presented in the respective column. **e** Proportion of the lost and gained differential interactions between TAMR and MCF7 cells that overlap ER binding sites. **f** Gene Ontology terms enriched at ER-bound differential interactions gained in TAMR cells as compared to MCF7 cells.

interactions at these altered ER-bound regions. In TAMR cells hypomethylated DMRs are strongly enriched at ER binding sites gained in TAMR cells (Fig. 4a). Conversely regions that gain methylation in TAMR cells are enriched at ER binding sites that are lost in TAMR cells as compared to MCF7 (Fig. 4a). To further confirm this relationship between aberrant ER binding and DNA

methylation, we plotted an average methylation profile around gained and lost ER binding sites in TAMR and MCF7 cells (Fig. 4b). We observed a genome-wide decrease in methylation in TAMR cells at ER sites that were gained, while loss of ER binding was associated with increased methylation (Fig. 4b, left panel). The opposite association was observed around gained and lost ER

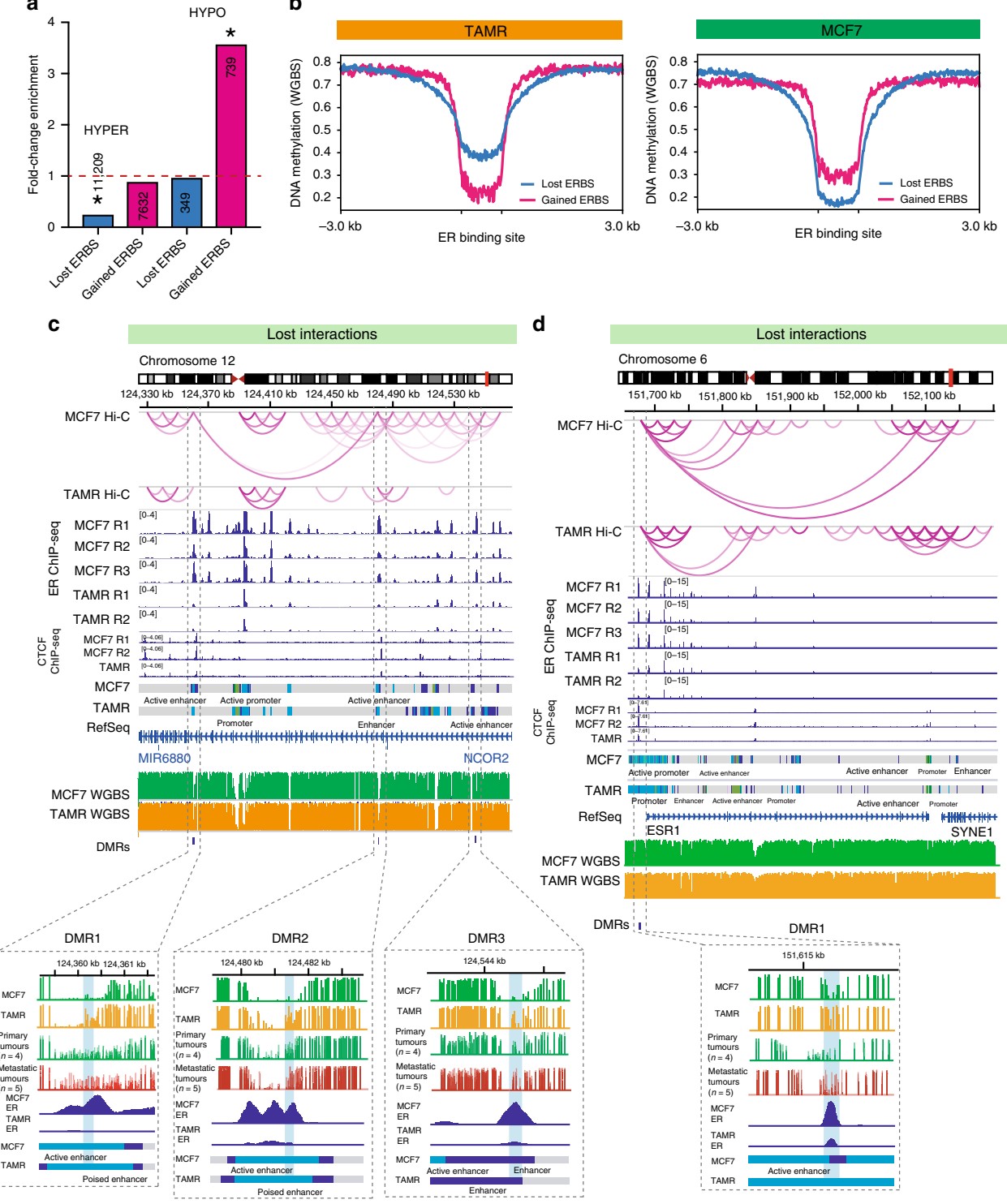

**Fig. 4 Methylation associates with loss of ER binding and loss of interactions. a** Fold-change enrichment of differentially methylated regions (DMRs), which are hyper- or hypomethylated in TAMR cells as compared to MCF7 cells for ER binding sites (ERBS). *$P$ value < 0.001. **b** Average profile plot showing methylation in TAMR (left panel) and MCF7 (right panel) around ER binding sites (ERBS) gained (pink) or lost (blue) in TAMR cells as compared to MCF7 cells. **c** Representative example showing loss of ER-bound interaction in TAMR cells at *NCOR2* gene associated with DNA hypermethylation. Three differentially methylated regions (DMRs) that are present at ER-enhancer regions are associated with loss of ER binding and loss of interactions in TAMR cells. DNA hypermethylation at the anchors of lost differential interaction, overlapping region of ER binding in MCF7 cells can be observed in metastatic ER+ breast cancer patient tumour samples ($n = 5$) as compared to primary tumours ($n = 4$). **d** Representative example showing loss of ER-bound interaction in TAMR cells at *ESR1* gene associated with DNA hypermethylation. Differentially methylated region (DMRs) at the ER-enhancer is associated with loss of ER binding and loss of interactions in TAMR cells. DNA hypermethylation at the anchor of lost differential interaction, overlapping a region of ER binding in MCF7 cells can be observed in metastatic ER+ breast cancer patient tumour samples ($n = 5$) as compared to primary tumours ($n = 4$).

binding and methylation in MCF7 cells (Fig. 4b, right panel). We then investigated if the association between DNA methylation and ER binding is maintained in clinical samples. Using ER ChIP-seq binding in primary breast cancers from patients with different clinical outcomes (nine "Responders" vs. nine "Non-responders" and three "Metastatic tumours"), we identified ER binding sites, which were lost in non-responders ($n = 14,553$) and gained in non-responders ($n = 1662$). We then examined the DNA methylation profiles from primary (i.e. endocrine-sensitive) and metastatic (i.e. endocrine-resistant) breast tumours around lost and gained ER binding sites that were located at the ERE motif. We observed a significant gain of methylation in metastatic samples at ER binding sites lost in non-responders (Mann−Whitney $P$ value = 0.0106) and an overall loss of methylation at gained ER binding sites (not significant) (Supplementary Fig. 4c).

Next, we interrogated our Hi-C data to determine whether altered ER binding found at regions of DNA hypermethylation leads to altered chromatin interactions in endocrine-resistant cells. Figure 4c, d and Supplementary Fig. 4g show representative examples of ER-bound differential enhancer−promoter interactions between TAMR and MCF7 cells, associated with altered ER-enhancer DNA methylation. For example, the enhancer region of the *NCOR2* gene, which gains methylation in TAMR cells, shows concomitant loss of ER binding and altered chromatin interactions at this region (Fig. 4c). Hypermethylation at the ER-enhancer regions where chromatin interactions are lost in the TAMRs relative to MCF7 cells can be observed in a zoomed-in view of each of the identified DMRs. A gain of methylation is also observed at some of these loci in ER+ endocrine-resistant clinical tumour samples ($n = 4$) as compared to matched primary tumours from the same patients ($n = 4$) (Fig. 4c and Supplementary Fig. 4d). Similarly, the *ESR1* (Fig. 4d and Supplementary Fig. 4e) and *MSI2* (Supplementary Fig. 4f−h) gene loci exemplify a hypermethylated DMR, which is associated with loss of local interactions and located at an ER-bound enhancer region in TAMRs and ER+ endocrine-resistant clinical tumour samples as compared to their matched primary tumours. However, there is large variability in DNA methylation observed between patient samples at each of the CpG sites within the shown DMRs. In some cases where there is hypermethylation in the primary tumour, intrinsic or de novo methylation at these sites may be suggested, which impacts the susceptibility to acquire endocrine resistance[6,36]. These data support that DNA hypermethylation we observe in endocrine-resistant cells, including clinical samples, is associated with loss of ER binding and differential chromatin interactions.

**Loss of TAD boundaries associates with loss of CTCF insulation**. We next investigated if a higher level of chromatin 3D architecture was also altered in endocrine resistance. We identified 2633 TADs with a median size of 720 kb in MCF7 cells. In endocrine-resistant cells, we identified 2641 TADs with a median size of 760 kb in TAMR cells and 2234 TADs with a median size of 800 kb in FASR cells (Supplementary Fig. 5a). Interestingly, the development of endocrine resistance led to significant increase in TAD size in FASR cells (Student's $t$ test $P < 0.0001$), but not in TAMR cells (Supplementary Fig. 5a). To quantify this genome-wide, we analysed the correlation between directionality index scores per 40 kb bins between TAMR, FASR and parental MCF7 cells. We observed a high overall correlation (Pearson's $R > 0.9$, $P$ value < 0.001) between MCF7 and both TAMR and FASR cells; however, FASR cells showed more variance in directionality index scores compared to MCF7, suggesting larger changes in domain organisation (Supplementary Fig. 5b, c).

We also investigated TAD boundary strength across the three cell types. Around 10% (between 10.5 and 12.8%) of identified significant intra-chromosomal interactions (at 40 kb resolution) crossed TAD boundaries, which is significantly below that expected at random for all three cell types (ANOVA $P$ value < 0.0001) (Supplementary Fig. 5d). The frequency of TAD boundary crossing was similar between MCF7 and TAMR cells and increased in FASR cells compared to random (not significant). Approximately 16% of TAD boundaries changed in tamoxifen resistance, with 828 TAD boundaries lost and 844 gained in TAMR cells (Supplementary Fig. 5e). In fulvestrant resistance, around 17% of TAD boundaries changed, with 1536 MCF7 boundaries lost in FASR cells and 738 ectopic FASR boundaries gained (Supplementary Fig. 5f). Interestingly, while we found that there was a ~34.2% overlap of new TAD boundaries created in both TAMR and FASR cells (Fig. 5a), there was ~77.8% overlap in the TAD boundaries that were lost in both TAMR and FASR cells as compared to MCF7 (Fig. 5b).

Generally we found that all stable and altered TAD boundaries displayed known characteristics of TAD organisation, with significant enrichment for CTCF binding, active promoter and enhancer histone marks (H3K4me3, H2AZac and H3K27ac) and no significant enrichment of polycomb-repressed histone mark (H3K27me3) (Supplementary Fig. 5g). However, TAD boundaries that were commonly lost in both endocrine-resistant cell types showed reduced enrichment for CTCF binding, while CTCF occupancy in MCF7 cells was enriched at these regions (Fig. 5c). Moreover, commonly gained TAD boundaries showed significantly higher enrichment for CTCF binding in TAMR and FASR cells, while in MCF7 cells CTCF binding was significantly depleted at these regions (Fig. 5c). This suggests that redistribution of CTCF may contribute to alterations in TAD boundaries in endocrine-resistant cell lines. An example is presented on chromosome 3 (Fig. 5d and Supplementary Fig. 5h), where in MCF7 cells this region is organised into three TADs and CTCF binding is significantly enriched at TAD boundaries compared to within the domains. The TAD boundary is lost in both TAMR and FASR cells and is associated with loss of CTCF binding at the region where two TADs are merged. This change can be observed in all replicates of the Hi-C data (Supplementary Fig. 5h). An example of a TAD boundary, which is gained only in FASRs, can be observed at a region on chromosome 4 (Fig. 5e and Supplementary Fig. 5i). This region is segmented into three distinctive TADs in MCF7 cells; however, in FASR cells this region is segmented into five sub-TADs. ChIP-seq binding shows large clusters of CTCF binding sites positioned at regions where TAD boundaries are present and increased CTCF binding is associated with the ectopic TAD boundaries. Altogether, while we observed minimal changes in TAD architecture to be associated with development of endocrine resistance, those that changed were focussed at regions of altered CTCF binding.

**Compartment structure reflects expression and ER binding**. Hi-C data can also be analysed to segregate the genome into either A-type or B-type compartments, which are associated with open and closed chromatin respectively and differential gene expression[9,12,39]. We first asked if the development of endocrine resistance affects genome segmentation. Each genome was separated into 25 kb consecutive bins, with each bin marked as type A (active) or type B (inactive) compartments and we compared the compartment status of each bin between the MCF7 and TAMR and FASR cells. Overall, we observed a high correlation (Pearson $R > 0.8$, $P < 0.001$) of compartment eigenvector values (see Methods) between MCF7 and TAMR and MCF7 and FASR cells (Supplementary Fig. 6a). Next, we tested the relative levels of

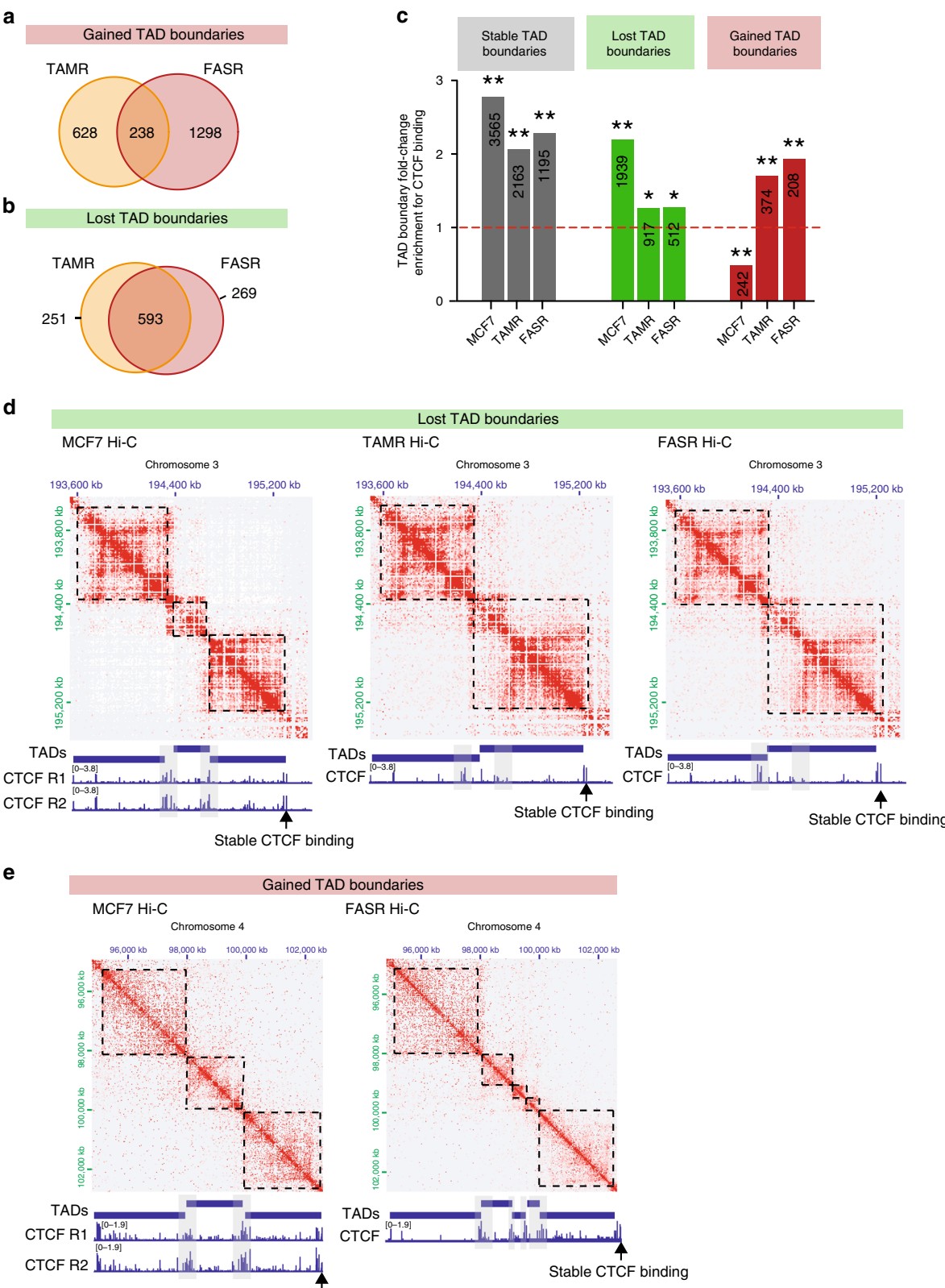

histone modifications across the A and B compartments for five different histone modifications/variants, including H3K27ac, H3K4me1, H3K4me3, H2AZac and H3K27me3 in MCF7, TAMR and FASR cells. We observed that the two compartment categories partition the active from inactive chromatin with A-type compartments showing high enrichment for "active" histone

modifications H3K4me3, H3K4me1 and H3K27ac and B-type compartments showing enrichment for "repressive" histone mark H3K27me3 in all three cell types (Fig. 6a and Supplementary Fig. 6b). Additionally, TADs identified in each of the cell types were mainly located within the compartment regions (68.21% in MCF7, 68% in TAMR and 60.65% in FASR) with most of the

**Fig. 5 Loss of TAD boundaries associates with decreased CTCF insulation. a** Overlap between TAD boundaries that were gained in TAMR and FASR cells as compared to MCF7 cells. **b** Overlap between TAD boundaries that were lost in TAMR and FASR cells as compared to MCF7 cells. **c** CTCF binding enrichment at stable and altered TAD boundaries, compared to random, distance-matched regions in three cell types. Asterisks represent the significance of fold-change enrichment at observed vs. random regions (permutation test **$P$ value < 0.001; *$P$ value < 0.05). The numbers of CTCF binding sites located within each specific region are presented in the respective column. **d** Representative example of a lost TAD boundary in endocrine-resistant cells. Hi-C interaction heatmaps visualised in JuiceBox for chromosome 3 for MCF7, TAMR and FASR cells aligned with CTCF ChIP-seq showing segmentation into TADs. Arrow marks a TAD boundary present in MCF7 cells and marked by high CTCF binding, which is lost in TAMR and FASR cells. Loss of TAD boundary is associated with loss of CTCF insulation at this region in TAMR and FASR cells. Per replicate data are shown in Supplementary Fig. 5h. Stable CTCF binding at TAD boundaries is marked by arrow as control loci. **e** Representative example of a gained TAD boundary in endocrine-resistant cells. Hi-C interaction heatmaps visualised in JuiceBox for chromosome 4 for MCF7 and FASR cells aligned with CTCF ChIP-seq showing segmentation into TADs. Arrow marks a region where three large domains in MCF7 cells are split into multiple sub-TADs in FASR cells. Ectopic TAD boundaries are associated with increased CTCF insulation at these regions in FASR cells. Per replicate data are shown in Supplementary Fig. 5i. Stable CTCF binding at TAD boundaries is marked by arrow as control loci.

compartment boundaries overlapping a TAD boundary (58.9% in MCF7, 57.8% in TAMR and 62.4% in FASR). The majority of compartments remained stable between MCF7, TAMR and FASR cells, with an average of 48.6% of the genome consisting of consecutive, "stable" A-compartments and 49.1% consisting of constitutive, "stable" B-compartments between MCF7 and TAMR cells (48.2% and 47.1% in FASR cells, respectively) (Fig. 6b). Upon tamoxifen resistance (MCF7 vs. TAMR cells), a total of ~1% of compartments switched its compartmentalisation from A-type to B-type and ~1.3% showed alteration from B-type to A-type (Fig. 6b). Upon fulvestrant resistance (MCF7 vs. FASR cells), a total of ~3% of compartments switched compartmentalisation from A-type to B-type and ~1.7% showed alteration from B-type to A-type (Fig. 6b). Most of the observed compartment switching was specific to the type of resistance acquired, with only four compartments identified that switched from A to B and four that switched from B to A in both FASR and TAMR cells.

Compartmentalisation of the genome has been previously reported to be associated with gene expression[12,17]. To better understand the link between compartment switching associated with endocrine resistance and gene expression changes, we investigated the respective log2 fold change RNA-seq expression levels of the genes that were located either at "stable" compartments ("A to A" and "B to B") or the compartments that switch their status ("A to B" and "B to A") between MCF7 and TAMR cells or between MCF7 and FASR cells (Fig. 6c). The genes located in compartments that switch from A-type to B-type with endocrine resistance showed significantly lower expression levels than genes located at regions that remained in the same compartment type ("stable") in both TAMR and FASR cells (Fig. 6c). In contrast, the genes located in regions that switched from B-type to A-type compartments in endocrine-resistant cells showed a pronounced gain in gene expression (Wilcoxon rank-sum test; $P = 1.74$E-05 [MCF7/TAMR] and $P = 6.01$E-11, [MCF7/FASR]) compared to genes located in "stable" compartments. Figure 6d shows an example of a compartment change from A-type to B-type, which occurs in both TAMR and FASR cells, and is associated with significant loss of gene expression in TAMR and FASR cells (Fig. 6d). Interestingly, decreased expression of *ATRNL1* (logrank $P = 0.0037$) and *GFRA1* (logrank $P = 0.081$) is associated with reduced overall survival in a cohort of ER+ breast cancer patients treated with endocrine therapy (Supplementary Fig. 6c, d)[24]. Further examples of compartment switching in either FASR or TAMR cells are presented in Supplementary Fig. 6e, f. Together these results demonstrate a significant correlation between gene expression and differential compartmentalisation upon endocrine resistance that is mostly specific to the type of resistance acquired.

We next plotted the average compartment eigenvector values around ER binding sites that changed in TAMR cells, as compared to MCF7 cells. Again we observed a strong cell-type-specific change in compartment organisation at gained and lost ER binding, suggesting that ER remodelling in endocrine resistance may be involved in switching between A and B compartments (Fig. 6e). To quantify this result genome-wide, we analysed the enrichment of different ER binding events at different compartment switching categories. Lost ER binding sites were highly enriched at compartments that switched from A-type to B-type. Similarly, gained ER binding sites were enriched at compartments that changed from B-type to A-type (Fig. 6f). Figure 6g exemplifies a region that switches from A-type in MCF7 cells to B-type in TAMR cells and this change coincides with loss of ER binding in TAMR cells at the *PTPRM* gene locus. This change in compartment structure is associated with increased expression of *PTPRM* in TAMR cells (Fig. 6h). High expression of *PTPRM* is also significantly associated with reduced relapse-free survival in ER+ breast cancer patients treated with endocrine therapy[24] (logrank $P = 0.025$) (Fig. 6h). An example of a B to A compartment switch that is associated with gain in ER binding in resistant cells is illustrated in Supplementary Fig. 6g. Together our results show a significant association between ER binding, compartment structure and gene expression and highlights the potential role of large-scale compartment changes in altered expression of ER-regulated genes with endocrine resistance.

## Discussion

Previous studies have shown that chromatin interactions differ between normal mammary epithelial and breast cancer cells[17] and ER binding is involved in long-range enhancer–promoter interactions[40]. Here we specifically ask how ER+ endocrine resistance affects the 3D chromatin organisation in breast cancer cells and what is the relationship with alterations in the genome and epigenome. Using in situ Hi-C we generated high-resolution 3D genome maps in two well-studied endocrine-resistant cell lines, FASR and TAMR, in comparison to the parental endocrine-sensitive cell line MCF7. Our multi-level analyses of 3D genome structure reveal alterations in long-range chromatin interactions (Fig. 7a), topologically associated domains (Fig. 7b) and finally A and B compartment profiles (Fig. 7c). We show that alterations to 3D genome architecture in the endocrine-resistant cells are enriched for active enhancers, active promoters and ER binding and, moreover, are frequently associated with epigenetic and genetic alterations, including hypermethylation or mutation of CTCF sites. These results provide evidence that 3D chromatin structure in ER+ breast cancer cells undergoes significant remodelling during development of endocrine resistance, regardless of the mode of endocrine treatment.

Maintenance of ER signalling is critical for the success of endocrine treatments and development of resistance is associated

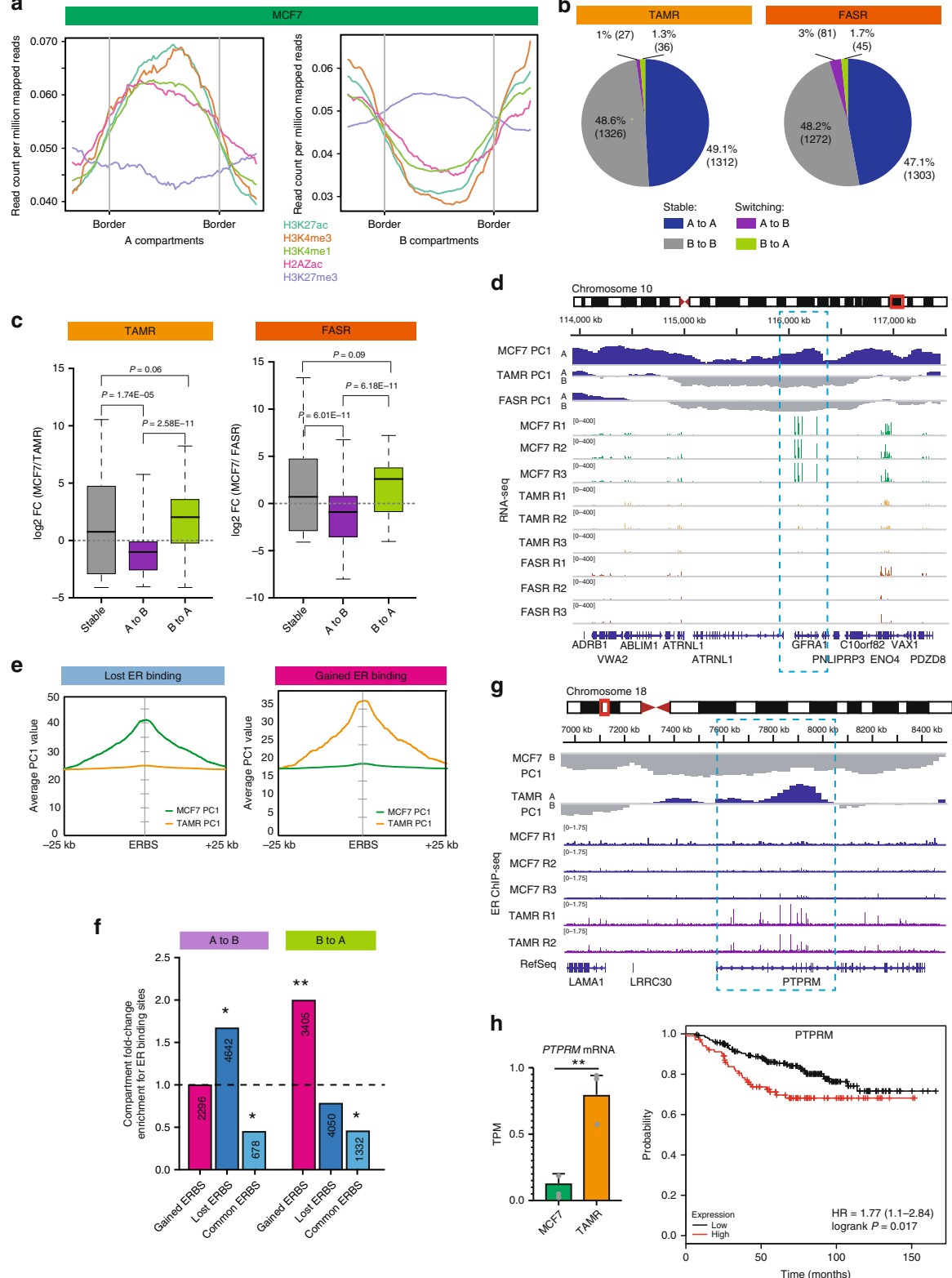

with loss of ER expression[36] or ER mutations[41]. Upon development of tamoxifen resistance in MCF7 cells (TAMR cells), we observed a significant loss of *ESR1* mRNA expression and reprogramming of ER binding sites across the genome. In fulvestrant-resistant cells (FASR), *ESR1* gene becomes silenced, with almost complete loss of expression. In our previous work, we found that hypermethylation at ER-responsive enhancers is an important mechanism of endocrine resistance[6]. Here, we suggest

that differential ER-bound enhancer−promoter interactions contribute to an underlying mechanism of endocrine resistance, where aberrant DNA methylation dynamically alters the ER-regulated enhancer−promoter interactions thus resulting in suppression of ER signalling pathways. Notably, ER-bound enhancer regions that gain methylation in TAMR cells are also frequently hypermethylated in endocrine-resistant tumour samples. High-resolution chromatin interactions maps allowed us to

**Fig. 6 Compartment structure reflects expression and ER binding. a** Average histone modification profiles over A and B compartments in MCF7 cells showing clear separation in active (A-type) and in-active (B-type) chromatin. **b** Pie chart showing the compartment changes in TAMR (left) and FASR (right) genomes as compared to MCF7 cells. "A" and "B" denote the open and closed compartments, respectively. "A to A" represents compartments that are open in both cell types, "B to B" represents compartments that are closed in both cell types, "A to B" denotes compartments that are open in MCF7 cells, but become closed in TAMR or FASR cells and "B to A" denotes compartments that are closed in MCF7 cells, but become open in TAMR or FASR cells. **c** Distribution of MCF7 vs. TAMR (left) and MCF7 vs. FASR (right) log2 fold change in gene expression for genes that change compartment status ("A to B" and "B to A") or remain within the same compartment type ("stable") (*P < 0.05). P value: Wilcox rank-sum test, SD. Source data are provided as a Source Data file. **d** An example of a region on chromosome 10 showing the compartment profiles at 25 kb resolution of parental MCF7 and endocrine-resistant TAMR and FASR cells, and MCF7 and FASR RNA-seq, showing a change in expression of genes located at regions that switch from A-type compartment status to B-type compartment status. **e** Histogram plot of average PC1 values around lost (left panel) and gained (right panel) ER binding sites (ERBS) in MCF7 and TAMR cells. **f** Different compartment switching regions between MCF7 and TAMR cells enrichment for common and unique ER binding compared to random, distance-matched regions. Asterisks represent the significance of fold-change enrichment at observed vs. random regions (permutation test) **P value < 0.001. *P value < 0.05. The numbers of ER binding sites located within each specific region are presented in the respective column. **g** An example of a region on chromosome 18 encompassing the *PTPRM* gene showing the compartment switching from "B-type" in parental MCF7 cells to "A-type" in endocrine-resistant TAMR cells that is associated with gain of ER binding in TAMR cells as compared to MCF7 cells. **h** *PTPRM* gene expression is lost in TAMR cells as compared to MCF7 cells (Student's *t* test P = 0.0021, SD, n = 3). Kaplan–Meier curves displaying relapse-free survival for 335 patients with ER+ tumours receiving endocrine treatment based on *PTPRM* gene expression. Patients with tumours with high expression of *PTPRM* are shown in red and those with low expression are shown in black. P value as indicated, log rank test. Source data are provided as a Source Data file.

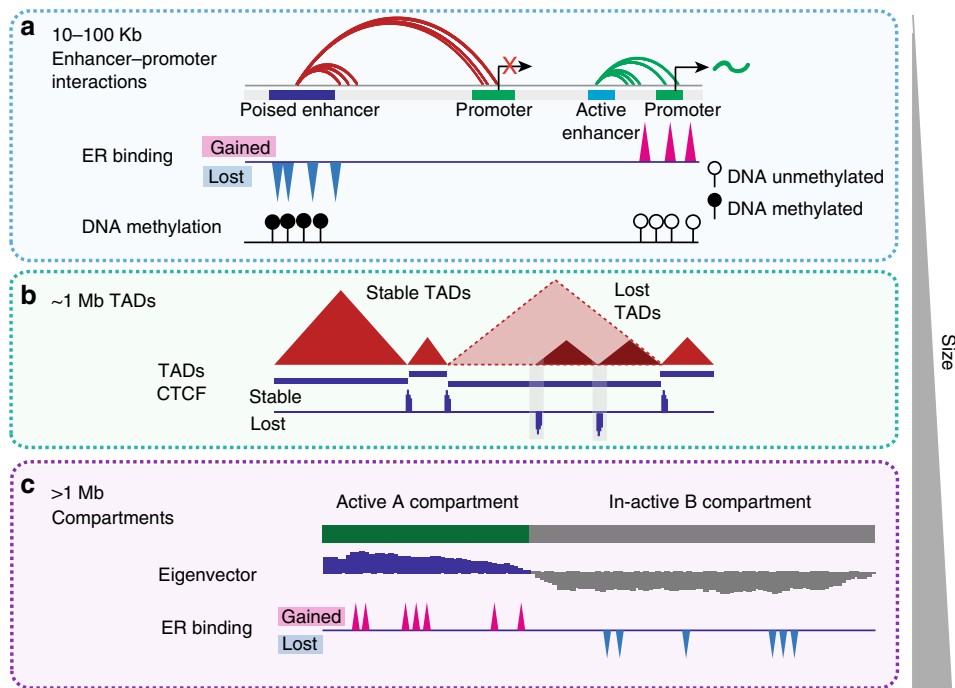

**Fig. 7 Model of 3D epigenome alterations in endocrine resistant breast cancer. a** Proposed model of epigenetic reprogramming in endocrine-resistant cells at the level of local enhancer–promoter interactions. Increased DNA methylation at ER-bound enhancer regions is associated with loss of ER binding and loss of ER-bound enhancer–promoter chromatin interactions, which results in decreased expression of ER-regulated genes. Gain of ER binding is associated with DNA hypomethylation and gain of ER-bound enhancer–promoter interactions and increased gene expression. **b** Proposed model of alterations of topologically associated domains (TADs) in endocrine-resistant cells. Loss or gain of CTCF insulation at TAD boundaries results in merging of adjacent domains or creation of new TAD boundaries. **c** Proposed model of epigenetic reprogramming in endocrine-resistant cells at the higher-level segmentation into A-type and B-type compartments. Regional reprogramming of ER binding is associated with altered compartment structure and long-range deregulation of gene expression.

show that this ER reprogramming in endocrine resistance is associated with rewiring of ER-bound interactions between active enhancers and promoters. Acquisition of novel ER-bound enhancer–promoter interactions is associated with aberrant expression of their target genes, with many of these target genes being involved in ER-signalling and coupled with patient's outcome. This observation expands on the previously suggested model in which ER is "pushed" to different sites in the genome during the development of endocrine resistance, which results in overall reprogramming of the ER binding landscape and aberrant gene activation[1]. Importantly, by investigating the underlying

genetic sequence in endocrine-resistant cells, we provide substantial evidence for the involvement of SNVs in the alteration of chromatin interactions and gene expression and provide a link between non-coding genetic variants and their target genes.

Topologically associating domains (TADs) are known to be common between multiple cell types; however, they can undergo significant alterations in cancer cells[20,21,42–44]. Comparison of TADs between MCF7 and endocrine-resistant cells revealed a minimal number of altered domains in the TAMR and FASR genomes (Fig. 7b). The two studied endocrine-resistant cell lines commonly lost TAD boundaries, while gain of new TAD

boundaries was more specific to each cell line, suggesting a common mechanism related to TAD boundary insulation is potentially altered in both resistant cell lines. Specifically, we found that TAD boundaries that are lost in endocrine-resistant cells were associated with a decreased CTCF insulation, demonstrated by loss of peaks at these TAD boundaries in resistant cells, while gain of TAD boundaries was enriched for ectopic CTCF binding (Fig. 7b).

Previous studies demonstrated that the genome is segmented into A and B compartments, containing active and repressed regions, respectively[9,45]. Currently, it is unclear how stable the arrangement of A/B compartments is across different cell types and in cancer. During stem cell differentiation, around 36% of the genome switched compartments and ~12% of compartments switched between epithelial (MCF10A) and breast cancer (MCF7) cells[12,17]. We found A/B compartments switching in endocrine-resistant cells was associated with differential expression of genes located within these regions and differential ER binding (Fig. 7c). Although in total only about 2–3% of all segments switched between A-type and B-type in TAMR and FASR genomes as compared to MCF7, these regions contained many important differentially expressed genes, including novel genes that have not been associated with endocrine resistance thus far. This finding provides further support for long-range coordinate regulation of gene expression previously shown in prostate[46,47] and breast cancers[48]. Finally, regions in the MCF7 genome, which switched from A-type to B-type compartment profile, showed a significant loss of ER binding sites, which provides strong support for the role of ER reprogramming in the control of chromatin interactions and gene expression.

Our work provides genome-wide evidence for the role of 3D genome structure and accompanying epigenetic, genetic and gene expression changes occurring in breast cancer endocrine resistance. However, further validation will be required to establish the direct link between 3D chromatin alterations observed in endocrine-resistant cell lines and specific changes that can be used as therapeutics targets or biomarkers in patients. Future experiments using genome editing to disrupt, delete or introduce TAD boundary elements or anchors of chromatin interactions will establish how identified 3D genome alterations influence the regulation of the corresponding genes in the context of endocrine-resistant breast cancer. Collectively, our results suggest a key epigenetic mechanism of endocrine resistance that is independent of the mode of therapy and offers a resource for further studies to explore future therapeutic applications.

## Methods

**Cell culture**. MCF7 breast cancer cells and the corresponding endocrine-resistant sub-cell lines were kindly given to our laboratory by Dr. Julia Gee (Cardiff University, UK). The MCF7 cells were maintained in RPMI-1640-based medium containing 5% (v/v) fetal calf serum (FCS). Tamoxifen-resistant MCF7 cells (TAMR) were previously generated by the long-term culture of MCF7 cells in phenol red-free RPMI medium containing 5% charcoal-stripped FCS and 4-OH-tamoxifen ($1 \times 10^{-7}$ M; TAM). FASR MCF7 cells were generated by the long-term culture of MCF7 cells in phenol red-free RPMI medium containing 5% charcoal-stripped FCS and fulvestrant ($1 \times 10^{-7}$ M; FAS). Endocrine resistance was characterised following >6 months of endocrine challenge exposure[15,16]. As a control, MCF7 cells (ATCC HTB-22) were also cultured across a 6-month time course in RPMI-1640-based medium containing 5% (v/v) FCS. All cell lines were authenticated by short-tandem repeat profiling (Cell Bank, Australia) and cultured for <6 months after authentication.

**Hi-C experiments**. Hi-C experiments were performed based on the in situ protocol by Rao et al.[49] with minor modifications. Single cells ($5–10 \times 10^6$ total) were collected and fixed with a final concentration of 1% formaldehyde for 10 min at room temperature with shaking. Reactions were quenched with glycine (0.125 M) and incubated for 5 min at room temperature, followed by 10 min incubation on ice. Cells were centrifuged for 10 min at $800 \times g$ at 4 °C, then washed in ice-cold phosphate buffered saline followed by an additional centrifugation. Nuclei were

extracted by incubation in 1 mL ice-cold lysis Buffer (10 mM Tris, pH 8.0, 10 mM NaCl and 0.2% NP-40 Igepal), supplemented with protease inhibitor cocktail for 2 h on ice with occasional mixing. Cells were then dounce homogenised by 30 slow strokes with pestle B. Nuclei were collected by centrifugation at 4 °C for 10 min and $2500 \times g$, then washed twice in ice-cold 1× NEB 3.1 buffer. Nuclei were re-suspended in 1× NEB 3.1 buffer with 10% SDS and incubated at 37 °C for exactly 60 min with shaking before adding 20% Triton X-100, re-suspending and incubating for another 60 min at 37 °C. Chromatin was digested overnight with 750U NcoIII restriction enzyme at 37 °C. DNA ends were repaired and marked with biotin-14-dATP (final concentration 28.4 μM) using Klenow Large Fragment DNA polymerase at 37 °C for 45 min. DNA was then centrifuged at $600 \times g$ for 6 min at 4 °C and supernatant was removed, leaving ~50 μL reaction volume including pellet. Ligation was performed in a final volume of 1000 μL, using 2000U of T4 Ligase, supplemented with 100 μL T4 DNA Ligase Buffer, 100 μL of 100% Triton X-100, 10 mg/mL BSA and nuclease-free water. Ligations were performed at 16 °C for 4 h prior to Proteinase K treatment overnight at 68 °C. DNA was purified twice with phenol chloroform isoamyl alcohol. After the second extraction, DNA was precipitated with 3 M sodium acetate and 100% ethanol overnight at −20 °C. DNA was collected by centrifugation at 18,000 × g for 20 min at 4 °C and washed twice with 80% ethanol. DNA was finally re-suspended in a final volume of 50 μL nuclease-free water.

**Hi-C library preparation and sequencing**. Hi-C libraries were prepared using NEBNext Nano II DNA Library Preparation kit for Illumina and customised protocol. Hi-C DNA was sonicated using a Covaris instrument to an average molecular weight of 300–500 bp. Fragmented DNA was repaired and blunt ends were dA-tailed using the NEBNext DNA Library Prep Master Mix Set for Illumina according to the manufacturers' instructions. A size selection was performed using AMPureXP Beads. Biotin-tagged DNA was bound to MyOne Streptavidin C1 beads using 2× Binding Buffer (10 mM Tris-HCl pH 8.0, 1 mM EDTA, 2 M NaCl) for 20 min at room temperature with rotation. Biotin-tagged DNA coupled with MyOne Streptavidin C1 beads was isolated using a magnetic particle concentrator. Beads were washed twice with 200 μL 1× Binding Buffer and once with 200 μL 1× Tween Wash Buffer (5 mM Tris-HCl pH 8.0, 0.5 mM EDTA, 1 M NaCl, 0.05% Tween). Beads were re-suspended in a final volume of 65 μL of water and adapters were ligated to DNA ends using the NEBNext Ultra II DNA Library Prep kit. PCR enrichment was performed using DNA bound to the MyOne Streptavidin C1 beads and NEBNext Multiplex Oligos for Illumina (Set 1 or 2) using the NEBNext Ultra II DNA Library Prep kit with eight cycles for library amplification. PCR products were purified using 1× volume of AMPure XP beads and eluted in 50 μL nuclease-free water. Hi-C libraries were then quantified using the KAPA Library Quantification Kit for Illumina and qualified using the Bioanalyzer 2100 (Agilent Technologies). Optimal concentrations to get the right cluster density were determined empirically. Resulting libraries were run on the HiSeq 2500 (Illumina) platform configured for 100 bp paired-end reads according to the manufacturer's instructions.

**Hi-C data processing**. We prepared three replicates of NcoI-digested in situ Hi-C libraries from parental, endocrine-sensitive ER+ MCF7 cells, tamoxifen-resistant TAMR cells and fulvestrant-resistant FASR cells. Raw Hi-C sequence data were mapped and processed using the HiC-Pro pipeline (version 2.9.0)[50]. Data were mapped to hg38/GRCh38. Statistics on the number of read pairs, uniquely mapped di-tags, valid interactions and interactions in cis per replicate are presented in Supplementary Table 1. Valid pairs files at 10 kb, 20 kb, 40 kb, 150 kb and 1 Mb resolution from HiC-Pro were transformed to JuiceBox-ready files using hic-pro2juicebox script from HiC-Pro. Multidimensional-scaling plots were constructed using plotMDS function applied to diffHiC processed filtered and normalised counts for each bin pair for each library. The distance between each pair of libraries is computed as the "leading log fold change", defined as the root-mean-square average of the top 1000 most variable bin pairs at 5 Mb, 1 Mb and 100 kb resolution. Interaction data were normalised using KR Observed/Expected; KR-normalised scores were visualised in WashU Epigenome Browser[51] or JuiceBox (version 1.6.2)[49,52].

**Detecting differential interactions (DIs)**. Differential interactions between parental MCF7 and endocrine-resistant TAMR and FASR cells were identified with Bioconductor diffHiC (v.1.6) R package[18]. Paired-end reads were aligned to hg38/GRCh38 with Bowtie2 (v.2.3.2), low-abundance reads were filtered out and the resulting data were normalised for direct, trended or CNV-driven biases. The statistical framework of the edgeR package was used on the final InteractionSet consisting of 145,109 regions to model the biological variability and to test for significance of identified differential genomic interactions. Differential interactions (DIs) were identified at 20 kb resolution with FDR cutoff of 5%. Differential interactions identified in TAMR/MCF7 diffHiC are presented in Supplementary Data 1 and FASR/MCF7 diffHiC in Supplementary Data 2.

**Detecting TAD boundaries**. Topologically associated domains were identified using a "domain-caller" pipeline developed by Bing Ren[10] in MATLAB (v. R2015b). The algorithm is based on the imbalance between the upstream and

downstream contacts of a region that is created by TAD. This imbalance is an indicator of whether a region is in the topological domain, at the boundary, or far away from a TAD and it can be quantified in a statistic called directionality index (DI). Domain-caller algorithm uses hidden Markov model (HMM) to determine the underlying bias state for each locus (upstream, downstream, none) and then these HMM calls are used to infer TADs as continuous stretches of downstream bias states followed by upstream bias states. For each cell type, a combined list of TADs was generated at 40 kb resolution and TAD boundaries were obtained by creating 10 kb flanks around start or end of each TAD. Similarly oriented boundaries within 40 kb from each other that were present in both cell lines were considered to be constitutive domain boundaries and remaining boundaries were considered to be cell-type-specific domain boundaries. The 40 kb window was chosen to mimic the uncertainty of the domain boundary position due to the 20 kb resolution of the domain calling as described previously[20,53].

**Detection of chromosomal A/B compartments**. To identify chromosomal compartments from Hi-C data, we performed eigenvalue decomposition of the Hi-C correlation matrix using the PCA analysis in Homer package (version 4.8)[54]. The resolution was set to 25 kb and the window size to 100 kb. Compartments were defined as regions of continuous positive or negative PC1 values using the find-HiCCompartments.pl tool in Homer. To detect which compartment is the open "A-type" and which is the closed "B-type," the genome-wide gene density was calculated to assign the "A-type" and "B-type" compartmentalisation. To identify genomic regions that switch between two compartment types, we used correlation difference tool (getHiCcorrDiff.pl) with findHiCCompartments.pl tool in Homer.

**ChIP-seq experiments and analysis**. ChIP-seq experiments for MCF7, TAMR and FASR cell lines were performed as previously described[55]. Antibodies used were H3K4me3 (Active Motif #39159), H3K4me1 (Active Motif, #39297), H3K27ac (Active Motif, #39133), H2AZac (Abcam, #ab18262), H3K27me3 (Millipore, #07-449) and CTCF (Millipore #07-729). ER ChIP-seq data for cell lines MCF7 and TAMR and clinical samples of ER+ breast cancer tumours were obtained from GEO (GSE32222)[37]. All ChIP-seq raw datasets were mapped and processed through the NGSane framework (v.0.5.2.0)[56]. Reads were mapped to genome build hg38/GRCh38 with bowtie v.1.1.0 and mismatched (>3 mismatched bases), multiple mapping and duplicate reads were excluded from downstream analysis. Chromatin modification peaks (H3K4me1, H3K4me3, H3K27ac, H2AZAC and H3K27me3) and transcription factor peaks (CTCF, ER) were identified using the Macs2 software (2.1.0) under the default parameters (band width = 300, model fold = [5, 50], qvalue cutoff = 5.00e-02). Consensus peaks were identified by intersecting Masc2 peaks obtained from each replicate using bedtools intersect (v.2.25.0). ChromHMM[19] (v.1.12) was applied to the chromatin modification-aligned reads to simultaneously partition the genome of each cell line into ten chromatin states. Binary files were created using BinarizeBam. In all, 8–15 models were created with LearnModel using default parameters. We chose the ten-state model as it displayed the most informative states while maintaining a manageable number of pairwise state transitions for interpretability. Pairwise combinations were counted per 200 bp bin (default bin size for ChromHMM) along the genome. DiffBind 2.4.8R package was used to perform differential binding analysis of ER ChIP-seq peaks between MCF7 and TAMR cell lines as well as between clinical samples from Responders (n = 9) and Non-responders (n = 9 and n = 3 Metastatic samples) (GSE32222)[37]. ChIP-seq binding profiles were generated using NGSane 0.5.2[56] or deepTools[57] and normalised to library size. Sequencing, mapping and peak calling information is provided in Supplementary Data 6.

**Whole genome sequencing experiments**. DNA was extracted from MCF7, TAMR and FASR cells and libraries prepared using the Nextera TruSeq Illumina kit. Genomic DNA was sheared, end repaired, ligated with barcoded Illumina sequencing adapters, amplified and size selected. Resulting Illumina sequencing libraries were then qPCR quantified, pooled and sequenced with 150 bp paired-end reads using Illumina HiSeq X Ten sequencers. Mean coverage across all samples was 34.59-fold (range 26.77–38.56).

**Single nucleotide variant analysis**. Sequenced reads were aligned to human reference genome hg38 using bwa-mem v.0.7.9[58] and the GATK pipeline v.3.5 was used to identify duplicate reads, perform local re-alignment at indels, base quality score recalibration. Mapping statistics were calculated using QualiMap v.2.1.3[59]. The somatic mutations (single nucleotide variants) were called between parental MCF7 and endocrine-resistant TAMR or FASR pairs with MuTect2 as previously described[60] using default parameters. In MuTect2 analysis all variant sites present in the dbSNP v.146 resource and COSMIC v79 coding and non-coding variants are used to aid filtering of known germline variants, leaving somatic and unspecified variants. MuTect2 pipeline does not automatically filter-out dbSNP variants and only uses the dbSNP overlap to require more evidence through a more strict LOD threshold (–dbsnp_normal_lod 5.5). VCF files generated by MuTect2 were further filtered using vcflib (https://github.com/vcflib/vcflib). SNVs that were marked with "PASS" in the MuTect2 output file were included in further analysis of somatic endocrine-resistance-associated variation. SNVs that have been previously identified in dbSNP v.146 were included if they fulfilled all the requirements of MuTect2

to be called somatic[60], including that they were not previously reported as germline (dbSNP SAO flag "germline"). Information on identified single-nucleotide variants in each of the resistant cell line is provided in Supplementary Table 2.

**Whole genome bisulphite sequencing experiments and analysis**. Library preparation and indexing were carried out as described in the CEGX TruMethyl WG user guide v2 with minor improvements as described in Nair et al.[61]. Sequencing reads from WGBS data were aligned to the human genome using version 1.2 of an internally developed pipeline Meth10X (https://github.com/luuloi/Meth10X[61]). The pipeline backbone is built based on workflow control Bpipe[62] version 0.9.9.2 which mainly helps to make automation, parallelism, restarting jobs and integration with cluster resource managers easier following the pipeline structure of P3BSseq[63]. Meth10X takes raw bisulphite reads in fastq files as inputs and produces an html report of all necessary metrics of bisulphite quality control and a tsv file of DNA methylation of 28 million CpG as rows and samples as columns. Briefly, adaptor sequences were removed using Trim Galore v. 0.2.8 in paired-end mode following prep kit guide. Bwa-meth version 0.20 (https://github.com/brentp/bwa-meth) was then used to align reads to hg38 using bwa version 0.7.13 (https://github.com/lh3/bwa). The generated bam files are merged and marked duplication by Picard tools 2.3.0 (http://broadinstitute.github.io/picard). The merged bam files are checked with Qualimap 2.2.1[64] for quality as well as all metrics of WGBS. MethylDackel (https://github.com/dpryan79/MethylDackel) and Biscuit (https://github.com/zwdzwd/biscuit) are used to call DNA methylation and SNP. Samtools version 1.2[65] is used to manipulate bam files.

**Clinical sample acquisition**. All specimens were obtained from SJHC pathology and were de-identified FFPE tissue blocks. The studies were approved by John Wayne Cancer Institute IRB and an external IRB, WIRB USA. Specimens were coded and blinded to the individuals running the assays.

**RNA-seq experiments and analysis**. RNA was isolated from MCF7, TAMR and FASR cells at ~80% confluence using Trizol. The poly(A)-selected RNA libraries were prepared using Illumina TruSeq RNA Sample Preparation kit, spiked-in with ERCC controls and paired-end sequencing was performed using HiSeq2500 instrument. 100 bp paired-end reads for MCF7, TAMR and FASR cell lines in biological triplicate were processed using Trim Galore (version 0.11.2) for adapter trimming (parameter settings:–fastqc–paired_retain_unpaired–length 16) and STAR (version 2.4.0j) for mapping reads to the hg38/GRCh38 human genome build with GENCODE v21 used as a reference transcriptome (parameter settings:–quantMode TranscriptomeSAM–outFilterMatchNmin 101). Differential gene expression (DEG) analysis was performed using edgeR 3.18.1[66,67] and TMM normalisation was applied to normalise for RNA composition[68]. Genes with logFC > 4 and FDR < 0.05. RNA-seq tracks were generated using bedtools v.2.22[69] genomeCoverageBed to create normalised.bedGraph files and bedGraphToBigWig (USCS utils) to create.bigwig files.

**Motif analysis**. To assess the enrichment of transcription factor binding motifs in the anchors of differential interactions, we used "findMotifsGenome.pl" function of the HOMER package (v.4.7) at FDR < 10%. Enrichment was obtained by comparing to matched, randomised regions.

**Statistical tests**. Enrichment observed/expected analyses were performed using Genome Association Tester (GAT v.1.3.5[70]) with n = 1000 permutations. Fold-change enrichment was estimated by comparing to a set of randomly generated matched background regions. Z scores were calculated for each genomic annotation using R package genomation v.1.8.0[71]. Correlation analyses were performed using R utility *cor.test*. DeepTools 2.1.0[57] was used to plot average profiles for ChIP-seq binding. The Mann–Whitney–Wilcoxon tests and chi-square test were used for two-group non-parametric comparisons.

**Reporting summary**. Further information on research design is available in the Nature Research Reporting Summary linked to this article.

## Data availability

All datasets used in this study are summarised in Supplementary Table 3. Raw and processed Hi-C, ChIP-seq, WGBS, WGS, RNA-seq and ChIP-seq data that support the findings of this study have been deposited in the NCBI Gene Expression Omnibus (GEO) with the primary accession codes GSE118716 and GSE130916. All other relevant data supporting the key findings of this study are available within the article and its Supplementary Information files or from the corresponding author upon reasonable request. The source data underlying Figs. 1d, e, 3b, 6c, h and Supplementary Figs. 1d, e, 2, 3f, 4c–e, h, 5a, g are provided as a Source Data file. A reporting summary for this Article is available as a Supplementary Information file.

## Code availability

All software used is published and/or in the public domain. All pipelines and R scripts used in the study are available at https://github.com/JoannaAch/MS_2019.

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

## Acknowledgements

We thank members of the Clark Laboratory for helpful discussions and careful reading of the manuscript. This work was supported by the Kay Stubbs Cancer Council NSW Project Grant RG 16-02, Stand Up to Cancer—VARI Epigenetics Dream Team, the National Health and Medical Research Council (NHMRC) Project Grant #1128916 and National Breast Cancer Foundation Innovator Grant #IN-17-043. S.J.C. is a NHMRC Senior Principal Research Fellow #1063559, F.V.M. is an NBCF Postdoctoral Fellow #PF-11-13. D.S.H. is supported by a FFANY Breast grant (DSH). J.M.W.G is supported by a Breast Cancer Now Fellowship and the Tenovus Cancer Charity. This work was supported by computational resources provided by the Australian Government through NCI Raijin under the National Computational Merit Allocation Scheme.

## Author contributions

S.J.C. conceived and coordinated the overall study and wrote the manuscript. J.A.-K. designed and performed the experiments, performed the bioinformatics analysis and wrote the manuscript with contribution from all other authors. F.V.M. performed the Hi-C and ChIP-seq experiments with technical help from J.A-K., S.S., K.F.F., G.C.S., N.S. Y.-T. Q.D. helped in bioinformatics analysis and interpretation of the data. K.A.G. and C.M.G. produced and processed the RNA-seq data, respectively. W.Q., W.J.L. and S.N. prepared the libraries and sequencing of WGBS samples and P.L.L. analysed the data. C.E.C. performed the WGS experiments and J.M.W.G. provided the cell lines. D.S.H. and I.R.R. provided clinical specimens and data. C.S. contributed to interpretation of data and reviewed the manuscript.

## Competing interests

The authors declare no competing interests.

## Additional information

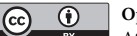

