## [Peer Review File · Nature Communications]

Reviewers' comments:

Reviewer #1 (Remarks to the Author):

In this paper, the authors make a series of comparisons using differential Hi-C interactions (chromatin state, gene expression, CTCF ChIP-seq peaks, Transcription factor binding sites and methylation of CpG sites) in endocrine sensitive MCF7 cells vs endocrine resistant sub-cell lines (TAMR and FASR).

I have one really fundamental concern regarding this paper; I don't think the authors have included adequate controls for their analyses. They make mechanistic inferences from their comparisons ie this change is due to endocrine resistance either generally, or specifically in response to Tamoxifen or Fulvestrant. However, these differences depend on comparing one cell line (MCF7) with two sub-cell lines (TAMR or FASR).

As I understand it all cell lines are heterogenous and growing any cell line in culture for 6 months (which I think is what they have done for the endocrine resistant cells) will result in changes to the overall population of cells. Specifically, Methods, cell culture, line 518 states "Endocrine-resistance was characterized following 6 months of endocrine challenge exposure (refs)."

First, what are the missing "refs"?

But more importantly, if the authors want to conclude that differences between the TAMR cells and the MCF7 cells are due to the endocrine resistance per se then they also need to grow the parent MCF7 cells for 6 months in culture without exposing them to Tamoxifen, and show that the TAMR cells are "more different" from the parental MCF7 cells than the long-term cultured MCF7 cells are from the parental MCF7 cells.

In the absence of this type of comparison it is not clear which differences are due to endocrine resistance, which differences are due to being long-term cultured and which differences are just random drift.

A second major concern relates to Results, page 6 line 167 onwards: "By integrating single-nucleotide variation (SNV) data obtained from Whole-Genome Sequencing (WGS) (see Methods) with differential interactions identified in TAMR and FASR cells, we investigated whether SNVs uniquely present in endocrine resistant cells, could directly alter individual differential 3D interactions."

There seems to be a fundamental confusion between somatic mutations and polymorphic germline variants. I am not familiar with MuTect2 but, as far as I can tell, it calls somatic point mutations. However, the variants that have been called by the authors (at least the ones in Fig 2C) are polymorphic germline variants (rs7180979, rs201722399 and rs113681683) one of which is quite common and two of which are extremely rare.

What exactly are the authors looking for here? Somatic mutations or polymorphic variants?

If they are looking for somatic mutations, why have they included polymorphic germline variants? And what proportion of the "enrichment" they find (Fig 2A) is due to polymorphic germline variants and what proportion is due to bona fide somatic mutations?

If they are looking for germline polymorphic variants, why would they expect them to have different alleles (or show differing levels of heterozygosity?) between the parental MCF7 cells and the sub-cell lines?

More minor comments

1. Page 4, line 122 onwards: GREB1 appears to be a gene that is down regulated at "lost interactions" in both TAMR vs MCF7 and FASR vs MCF7 (Fig 1D and Sup Fig 1A). However, the "lost interactions" are only shown for FASR (not TAMR). Are the same interactions lost in TAMR?

2. Page 5 line 157 -158 and "looping figures": the gene tracks in the looping figures are generally very unclear – for example in Sup Figs 1D and 1E it's impossible to isolate individual genes and in Sup Fig 1F, only one of seven ESR1 isoforms is shown. I think, from looking in UCSC that the loops that are maintained in TAMR cells show interactions between the various ESR1 promoters – do they?

And the loops that are lost, are between the ESR1 promoter and 3' end of ESR1. Why do the authors think that this is an example of a promoter enhancer interaction (page 5 line 157 -158)? This certainly isn't supported by the enhancer and promoter marks that they show in the same figure – the labelled "promoter marks" colocalise with the 3' end of ESR1 (not the 5' end).

Similarly, for GREB1 (Figure 1F), the long range interactions that are being lost don't appear to map to the GREB1 promoter? They appear to map upstream of GREB1 at ~11,480 kb and interact with regions at ~11,580kb (within introns 6 and 7).

Finally, could they add CTCF ChIP-seq tracks to these looping figures so that the reader can whether the long range looping interactions are colocalising with CTCF binding sites?

3. Page 7, line 207 onwards: "We found that in tamoxifen-resistance, interactions lost in TAMR cells were mainly present at ERE, SOX2 and FOXA1 binding sites, while interactions gained in TAMR were present at ZNF143, OCT4, RUNX2 and HOX cluster (HOXB13 and HOXD13) binding sites (Fig. 3A)."

What does "mainly present" mean? Compared to what?

Do they have any data to suggest that the factors actually bind to these sites (ie ChIP-seq data)? They access ChIP-seq data for ER in the next section (Fig 3D) so does this confirm these in silico predictions? Or not?

What other transcription factor binding sites did they look at that weren't enriched at anchor points that were lost or gained? How did they select the group of transcription factors that they tested? And what was the fold enrichment for the ones they list?

Are all of these transcription factors expressed in MCF7, TAMR and FASR cells?

4. Page 8, line 232 onwards: "Differential interactions gained in TAMR were highly enriched for gained ER binding, and depleted for lost ER binding sites and interactions lost in TAMR were enriched for lost ER binding events (Fig. 3D). Both lost and gained differential interactions showed a significant enrichment at common ER binding sites between MCF7 and TAMR cells (Fig. 3D).

The P-values in this figure are puzzling; they state that gained DIs are highly enriched for gained ER binding (although this isn't one of the binding sites listed as present at gained DIs in Fig 3A) and yet the P-value for this comparison is less significant than that for common ER binding sites (where the enrichment appears to be much less). Presumably this is because there is a large fold change in binding based on small amounts of absolute data? I think rather than just bar charts the authors need to present actual numbers of binding sites for each category and explain in detail how they selected their random regions (there is nothing in the Methods).

5. Page 9, line 280 onwards: "Similarly, the ESR1 gene locus exemplifies a hypermethylated DMR, which is associated with loss of local interactions and located at an ER-bound enhancer region in TAMRs and ER+ endocrine resistant clinical tumour samples (n=5) (Fig. 4D).

The implication of this statement is that this region would not be hypermethylated in ER+

endocrine sensitive clinical tumour samples – have the authors checked whether this is the case?

Reviewer #2 (Remarks to the Author):

The aim of the manuscript is to address the role of the 3D-chromatin structure in endocrine sensitive and endocrine resistant breast cancer cell lines with further consideration of epigenetic alterations. The manuscript is well written and the scientific question on identifying how ER+ endocrine resistance affects the 3D architecture is interesting and relevant. The approach of combining data sets of not only different samples, but different kind of data reaching from Hi-C, to RNA-seq, and ChIP-seq (and more) offers a great opportunity for in-depth data analysis. However, I am missing the results that clearly show that the 3D architecture is the key mechanism. I would have expected a suggestion of specific interactions or TADs in combination with relevant genes characterizing the resistance of the two cell lines more specifically. Moreover, it would be of interest to add an experiment or at least suggest it in the discussion, where important interactions or TADs are mutated (either in the TAMR and FASR cell lines to correct them, or in the MCF7 cell line to introduce them) to see how this influences the regulation of the corresponding genes in the context of endocrine resistant breast cancer. Another (future) possibility would be also to target the genes affected by interaction/TAD changes using genome editing.

Major comments:

1. Three cell lines were used, whereby TAMR and FASR were established from the ER+ MCF7 cell line. The TAMR (tamoxifen resistant) and FASR (fulvestrant resistant) cell lines should only be characterized by the difference characterized by their drug resistance. Thus, it would be helpful to show the RNA-seq data and provide a confirmation that the genes differentially expressed between MCF7 and TAMR, and MCF7 and FASR are associated with the corresponding genes characterizing them.

2. According to Figure 1 three Hi-C libraries were generated for each cell line. However, supplementary table 1 only shows one number per cell line. Thus, it is unclear if the presented number is the mean of each library, or if only one sample was used, or if it is the number of reads combined from all samples.

Depending on what is presented, samples should be either further sequenced to achieve a higher number of uniquely mapped reads or a replicate should be added for each cell line. According to the current literature >100 million unique valid pairs are recommended for a replicate (PMID: 22652625).

3. It seems that the FASR Hi-C sample has the lowest sequencing depth and the smallest number of unique valid pairs. Based on your conclusion that FASR has less interactions compared to TAMR and MCF7, the question comes up whether this is a true biological consequence due to fulvestrant resistance or if a subset of interaction couldn't be identified due to the lower sequencing depth. Thus, these samples should be sequenced to a more similar depth or the other two samples should be downsampled.

4. Adding ChIP-seq data to further characterize lost and gained interactions definitely brings in an additional layer of information. Thus, I would recommend to additionally show ChIP-seq profiles (rather than a bar plot as used in Figure 1C).

5. Figure 1D and 1E: Please provide the genes that are associated with lost and gained interactions along with their expression values as a suppl. table (RNA-seq data).

6. Figure 1F and 1H, along with the according suppl. figures: Please add the expression levels

(from RNA-seq data) to the shown genes. How many folds are they increased/decreased based on RNA-seq?

Furthermore, it can't be derived from the figure how the shown RNA-seq reads were normalized nor the track height. Please add this information to allow the comparison of the samples.

In general, it would be good to show the fold changes and expression values from the RNA-seq data and provide some ChIP-seq tracks to see if epigenetic changes occur in those regions.

7. As far as I know whole genome sequencing of cell lines is very tricky and it seems to me difficult to distinguish pre-existing SNV (those which are present in the cell line, but not related to resistance) from gained ones. The software you used for the analysis suggests to my understanding the usage of matched tumor and normal samples. Thus, it would be helpful if you describe your approach more detailed, but also how you identified the SNVs specific to TAMR or FASR and additional characteristics. How many SNVs did you identify? How many were shared, or specific ones? How did you handle SNV identified in repetitive and black regions of the genome? How many of the SNVs you found were known in dbSNP or other databases and in which context were the known ones identified? Finally, also which genes were affected due to SNVs associated with changed interactions.

8. You specifically looked at CTCF. How many loop anchors have CTCF binding? Is it the main TF at your loop anchors or may the SNVs also affect other TF binding motifs, that are maybe also relevant in those samples?

9. The illustration of the missing loop in Figure 2E should be confirmed using replicates, and/or using a more similar sequencing depth.

10. Figure 3C: It is not clear if you had for all ChIP-seq samples replicates, nor how deep they were sequenced. If you used replicates, how did you handle the replicates for applying peak calling?

11. Figure 3D: Binding profiles of the ChIP-seq samples would add further information.

12. Figure 4C and 4D: The expression levels and fold changes of those genes should be shown. Besides ER, H3K27ac would be an informative epigenetic mark.

13. TADs gained and lost should be further characterized. What type of genes are in those TADs (lost and gained, also considering TAMR and FASR specific vs shared ones)? How is the gene expression changed within those TADs – any trends? Do chromatin marks change within those TADs? What kind of genes are affected and how? Is it possible to derive gene expression characteristics or a specific pattern regarding epigenetic changes?

14. Figures 5 D and E: The pattern of the Hi-C data looks similar. Thus, the usage of Hi-C replicates and/or adjustment of the sequencing depth would give the possibility to further confirm the current results.

15. Figures 5 D and E: Please add the track height for the ChIP-seq data as well as how they were normalized and an independent control locus.

16. How did you take the TAD boundaries identified in Figure 5 into account for identifying the active and inactive compartments?

17. Where are the compartments that change from active to inactive or vice versa located? Can they be associated with changing TADs or interactions? Or from the other perspective – can changed interactions or TADs be associated with switching compartments? What types of genes are within those switching compartments?

18. Are there switched compartments which are specific to TAMR or FASR, or are they changed in both? Are specific genes affected?

More general comments:

19. Figure 1H seems to be missing in the figure legend

20. In those figures where chromatin changes are shown in bar plots, binding profiles would add further information.

21. Please add all track heights to the figures to allow a better comparison and describe the normalization used.

22. Please add control loci for ChIP-seq data, as they help to see if the reduction in binding is specific to the shown locus.

23. Besides showing the RNA-seq reads, the RNA-seq expression data along with fold changes should be shown as tables and maybe mentioned in the text.

24. How deep were the other data sequenced? How many replicates were used? How were replicates handled (like peak calling for ChIP-seq; identified interactions; ...)?

25. Please also add the used post-hoc tests for statistical tests where multiple comparisons were done.

26. In line 519: the references seem to be missing.

We appreciate the reviewer's time and constructive comments. We have performed additional Hi-C experiments and provided new analyses and revised the Manuscript accordingly. We have also expanded the Methods section and provided additional Supplementary Tables and Figures. We feel that the new data and analyses have further strengthened our manuscript and have addressed all the reviewers concerns.

We have marked changes in the manuscript in blue.

Please find a summary of our responses below:

Reviewer #1 (Remarks to the Author):

In this paper, the authors make a series of comparisons using differential Hi-C interactions (chromatin state, gene expression, CTCF ChIP-seq peaks, Transcription factor binding sites and methylation of CpG sites) in endocrine sensitive MCF7 cells vs endocrine resistant sub-cell lines (TAMR and FASR).

Major Comments:

I have one really fundamental concern regarding this paper; I don't think the authors have included adequate controls for their analyses. They make mechanistic inferences from their comparisons ie this change is due to endocrine resistance either generally, or specifically in response to Tamoxifen or Fulvestrant. However, these differences depend on comparing one cell line (MCF7) with two sub-cell lines (TAMR or FASR).

As I understand it all cell lines are heterogenous and growing any cell line in culture for 6 months (which I think is what they have done for the endocrine resistant cells) will result in changes to the overall population of cells. Specifically, Methods, cell culture, line 518 states "Endocrine-resistance was characterized following 6 months of endocrine challenge exposure (refs)."

First, what are the missing "refs"?

Response:

We apologise the missing references that describe the characterization of endocrine resistant cell lines TAMRs and FASRs. These are Knowlden, J.M. et al. Elevated levels of epidermal growth factor receptor/c-erbB2 heterodimers mediate an autocrine growth regulatory pathway in tamoxifen-resistant MCF-7 cells. *Endocrinology* 144, 1032-1044 (2003) and McClelland, R.A. et al. Enhanced epidermal growth factor receptor signaling in MCF7 breast cancer cells after long-term culture in the presence of the pure antiestrogen ICI 182,780 (Faslodex). *Endocrinology* 142, 2776-2788 (2001) (reference 15 and 16) and have now been added on page 18.

But more importantly, if the authors want to conclude that differences between the TAMR cells and the MCF7 cells are due to the endocrine resistance per se then they also need to grow the parent MCF7 cells for 6 months in culture without exposing them to Tamoxifen, and show that the TAMR cells are "more different" from the parental MCF7 cells than the long-term cultured MCF7 cells are from the parental MCF7 cells.

In the absence of this type of comparison it is not clear which differences are due to endocrine

resistance, which differences are due to being long-term cultured and which differences are just random drift.

Response:

We acknowledge the reviewers suggestion of additional controls to support our conclusions. In order to determine if the 3D chromatin differences we observed in the endocrine resistant cells are ‘just a random drift’ with long-term culture we have now performed Hi-C experiments on MCF7 cells that we previously grew over a 6 month time course.

We have performed HiC experiments (in duplicate) at three growth time points – start of time course (T0), mid-point (T3, ~3 months) and late culture (T6, 6 months) and compared the Hi-C 3D interaction maps obtained between 1) the MCF7 cells (triplicates) that were originally used to derive the endocrine resistant cells (TAMR and FASR), 2) public MCF7 cells (Barutcu et al., 2015), and 3) the new long-term cultured MCF7 time-point series (duplicates). By comparing all the different Hi-C MCF7 datasets with the TAMR and FASR Hi-C data, at multiple resolutions (5Mb, 1Mb, 100kb) on PCA plots, the data clearly shows that TAMR/FASR cells are “more different” from the parental MCF7 cells than the long-term cultured MCF7 cells are from the parental MCF7 cells (**new Supplementary Figure 1a**).

This data supports our conclusion that long-term culturing of MCF7 cells results in minimal changes to chromatin conformation, whereas the development of endocrine-resistance results in major changes in 3D structure (**new Supplementary Figure 1a**). This data has now been added to the Results (page 4) and Methods section (pages 18 (tissue culture) and 20 (data analysis)).

A second major concern relates to Results, page 6 line 167 onwards: “By integrating single-nucleotide variation (SNV) data obtained from Whole-Genome Sequencing (WGS) (see Methods) with differential interactions identified in TAMR and FASR cells, we investigated whether SNVs uniquely present in endocrine resistant cells, could directly alter individual differential 3D interactions.

There seems to be a fundamental confusion between somatic mutations and polymorphic germline variants. I am not familiar with MuTect2 but, as far as I can tell, it calls somatic point mutations. However, the variants that have been called by the authors (at least the ones in Fig 2C) are polymorphic germline variants (rs7180979, rs201722399 and rs113681683) one of which is quite common and two of which are extremely rare.

What exactly are the authors looking for here? Somatic mutations or polymorphic variants? If they are looking for somatic mutations, why have they included polymorphic germline variants? And what proportion of the “enrichment” they find (Fig 2A) is due to polymorphic germline variants and what proportion is due to bona fide somatic mutations?

If they are looking for germline polymorphic variants, why would they expect them to have different alleles (or show differing levels of heterozygosity?) between the parental MCF7 cells and the sub-cell lines?

Response:

We apologise for the confusion and lack of clarity in this section. To comprehensively address this reviewer and reviewer 2 concerns about the SNV analysis (see Reviewer 2 Questions 7 & 8) we have now re-analysed the data to only report somatic SNVs and removed any germline data as we agree that this is not informative for this study (See new result section pages 4-8).

We reanalysed WGS data from MCF7, TAMR and FASR cells with Mutect2 using GATK pipeline with the following stringent parameters to filter-out dbSNP v.146, COSMIC v79 coding and non-coding variants (parameter settings: `--dbsnp dbsnp_146.hg38.vcf --cosmic CosmicCodingMuts.hg38.vcf --cosmic CosmicNonCodingVariants.hg38.vcf`) and obtained a list of endocrine resistance-associated somatic variants (or specific to resistant cell line and not present in control cell line) (total = 14652 in TAMR cells and 15381 in FASR cells)

Summary of somatic SNV calls obtained from Mutect2 has now been included in **new Supplementary Table 5**. Mutect2 is a recommended software for matched tumour/normal analysis and is suggested for cell line comparison analysis by GATK Best Practises. To better characterize the SNVs, we first gathered the coordinates of SNVs associated with endocrine-resistance (14,652 in TAMR and 15,381 in FASR) and used HOMER `annotatePeaks.pl` to find their location in relation to genes. Both in TAMR and FASR cells, ~60% of resistance-associated SNVs mapped to intergenic regions, ~35% to introns and ~1% to coding exons (182 in TAMR and 139 in FASR) (**new Supplementary Fig. 3a**). Therefore, in order to identify what regulatory elements may be affected by identified SNVs, we looked at enrichment at MCF7 DNase-Seq sites (ENCODE ENCSR000EPJ). We observed a significant enrichment of TAMR SNVs at DNase-Seq peaks compared to genome-wide background, but not in FASR cells (**new Supplementary Fig. 3c**).

Given the 20kb resolution of the differential interactions anchors and that we are intersecting tens of thousands of CTCF binding sites, we decided to focus specifically on CTCF bindings sites that are lost in TAMRs/FASRs as compared to MCF7s and which are located within anchors of differential interactions. This approach will assist us in identifying putative "key" CTCF sites that are involved in differential interactions. Using this approach we identified 28 SNVs in TAMRs and 46 in FASRs, **three** of which were located directly within the CTCF motif. In both TAMR and FASR cells, more resistance-associated SNVs were located at anchors of interactions that were lost in resistant cells, as compared to interactions that were gained (new **Fig. 2c**). One of SNVs located directly at the CTCF motif is rs201722399 located on chromosome 17. This variant was detected by Mutect2 using stringent parameters to exclude germline variants: `initial_tumor_lod=4.0 initial_normal_lod=0.5 tumor_lod=6.3 normal_lod=2.2 dbsnp_normal_lod=5.5`. Our high coverage WGS data clearly shows that it is not present in the control MCF7 cells (as seen on the **Rebuttal Figure 1** – 27/27 reads are "A"), but is present in resistant cells in (**Rebuttal Figure 1** below, 33/40 reads are "A", 1 is "C" and 6 are "G"). It is possible for SNPs in dbSNP to pass MuTect2 filters, especially if there is no evidence for the variant in the normal sample.

Rebuttal Figure 1. Mapped reads from Whole Genome Sequencing experiments in MCF7 (top) and TAMR (bottom) around the rs201722399 SNV.

Minor comments:

1. Page 4, line 122 onwards: *GREB1* appears to be a gene that is down regulated at “lost interactions” in both TAMR vs MCF7 and FASR vs MCF7 (Fig 1D and Sup Fig 1A). However, the “lost interactions” are only shown for FASR (not TAMR). Are the same interactions lost in TAMR?

Response:

Indeed *GREB1* is one of the most differentially expressed genes in both TAMR and FASR cells as compared to MCF7 and its expression is lost in both resistant cell types (**new Supplementary Fig. 2a**). We have now included the TAMR interactions in the main figure in addition to FASR interactions (**new Figure 1f**) to demonstrate that loss of long-range interactions at the *GREB1* locus occurs both in TAMR and FASR cells and is associated with loss of *GREB1* expression in these cells.

2. Page 5 line 157 -158 and “looping figures”: the gene tracks in the looping figures are generally very unclear – for example in Sup Figs 1D and 1E it’s impossible to isolate individual genes and in Sup Fig 1F, only one of seven ESR1 isoforms is shown. I think, from looking in UCSC that the loops that are maintained in TAMR cells show interactions between the various ESR1 promoters –

do they?

And the loops that are lost, are between the *ESR1* promoter and 3' end of *ESR1*. Why do the authors think that this is an example of a promoter enhancer interaction (page 5 line 157 -158)? This certainly isn't supported by the enhancer and promoter marks that they show in the same figure – the labelled “promoter marks” colocalise with the 3' end of *ESR1* (not the 5' end).

Response:

We have now modified the Supplementary Figures to try and isolate individual genes. Previously, as the visualised regions are large in size, we “collapsed” RefSeq gene tracks in most of presented example figures. In the **new Supplementary Figure 2d** (previously Supplementary Figure 1E), the visualised region is only 180Kb in size but includes a total of 14 genes. We have now modified this figure and expanded the gene track. This has improved the interpretation and identification of the putative gene affected by this change in interactions, which is *TAPBP22*. We have also highlighted the promoter region of *TAPBP22* and the interactions lost at this region. Also in the **new Supplementary Figure 2c** (previously Supplementary Figure 1D) both genes located within the differentially “looping” region are differentially expressed (as shown in **new Supplementary Figure 2c**).

We have also modified the **new Supplementary Figure 2e** (previously Supplementary Figure 1F) to show the same regions with all *ESR1* isoforms included. As correctly pointed out, stable interactions suggest a putative promoter-promoter interactions between different isoforms of *ESR1*, which are maintained in MCF7 and TAMR cells, while long-range interactions between 3' and 5' of *ESR1* are lost in TAMR cells. We have now included a zoomed-in view of chromHMM tracks at 3' end of *ESR1* showing Active Promoter marks around the TSS (~151,700kb) and zoomed-in view of 5' end of *ESR1* showing multiple enhancer marks at these region (mainly around ~152,090kb) (**new Supplementary Figure 2e**). Therefore, we concluded that lost interactions at this region are putative enhancer-promoter interactions. We have marked the labels of the chromHMM marks in bold to reflect this (**new Supplementary Figure 2e**).

Similarly, for *GREB1* (Figure 1F), the long range interactions that are being lost don't appear to map to the *GREB1* promoter? They appear to map upstream of *GREB1* at ~11,480 kb and interact with regions at ~11,580kb (within introns 6 and 7). Finally, could they add CTCF ChIP-seq tracks to these looping figures so that the reader can whether the long range looping interactions are colocalising with CTCF binding sites?

Response:

The Hi-C data shown on the figure is at 10kb resolution and as such these interaction anchors may be anywhere within a 10kb region. However, the Figure shows that multiple interactions are being lost within the *GREB1* gene locus and some of them indeed map upstream of the *GREB1* putative promoter. We have now added CTCF ChIP-seq tracks as suggested to all figures showing long-range interactions to improve interpretation of presented loops (**new Figure 1f**).

3. Page 7, line 207 onwards: “We found that in tamoxifen-resistance, interactions lost in TAMR cells were mainly present at ERE, SOX2 and FOXA1 binding sites, while interactions gained in TAMR were present at ZNF143, OCT4, RUNX2 and HOX cluster (HOXB13 and HOXD13) binding sites (Fig. 3A).”

What does “mainly present” mean? Compared to what?

Do they have any data to suggest that the factors actually bind to these sites (ie ChIP-seq data)?

They access ChIP-seq data for ER in the next section (Fig 3D) so does this confirm these *in silico* predictions? Or not?

What other transcription factor binding sites did they look at that weren't enriched at anchor points that were lost or gained? How did they select the group of transcription factors that they tested? And what was the fold enrichment for the ones they list?

Are all of these transcription factors expressed in MCF7, TAMR and FASR cells?

Response:

We apologise for the lack of clarity. In **Figure 3a** we presented results from *in silico* Homer motif analysis (as described in Methods page 24) using matched, randomized genomic regions as background. We have now explained this in text, figure legend and in Methods accordingly. In order to confirm the predicted motifs, we now assessed the enrichment of all transcription factors binding sites in MCF7 cells available in ReMap 2018 v.1.2 (total: 235 datasets, 85 TFs) at anchors of differential interactions (**new Supplementary Table 7**). We observed large number of significantly enriched TF at differential interactions (P value < 0.001 and FC > 2). ESR1, MYC, CTCF, NR2F1, FOXA1 and PgR were most commonly observed. Out of 22 TF motifs identified in *in silico* analyses, public ChIP-seq data was available for 7 TF. Out of these 6 (i.e. MYC, NRF2, ARTN2, FOXA1, ESR1, PgR) showed significant observed/expected enrichment for binding at anchors of differential interactions, validating our *in silico* findings. These results are now included in the main manuscript and in **new Supplementary Table 7**.

Additionally, for 7 TF that showed strong enrichment (P value < 0.001 and FC > 2) and were detected in *in silico* Homer motif analysis, we looked at the expression of genes encoding these factors. All genes encoding these TF are expressed in these cell lines, however no association between expression levels and presence at lost/gained interactions was observed (**Rebuttal Figure 2**).

Rebuttal Figure 2. Gene expression of transcription factors enriched at differential interactions

4. Page 8, line 232 onwards: “Differential interactions gained in TAMR were highly enriched for gained ER binding, and depleted for lost ER binding sites and interactions lost in TAMR were enriched for lost ER binding events (Fig. 3D). Both lost and gained differential interactions showed a significant enrichment at common ER binding sites between MCF7 and TAMR cells (Fig. 3D).

The P-values in this figure are puzzling; they state that gained DIs are highly enriched for gained ER binding (although this isn't one of the binding sites listed as present at gained DIs in Fig 3A) and yet the P-value for this comparison is less significant than that for common ER binding sites (where the enrichment appears to be much less). Presumably this is because there is a large fold change in binding based on small amounts of absolute data? I think rather than just bar charts the authors need to present actual numbers of binding sites for each category and explain in detail how

they selected their random regions (there is nothing in the Methods).

Response:

We have now added the actual numbers of ER bindings sites in each category to the bar chart in **new Figure 3d**. In the observed/expected analysis P-value is calculated based on the number of overlapping features in the data as compared to the number of overlapping features in the background dataset (*in silico* digested hg38 genome with *NcoI* enzyme) with 1000 permutations. Less significant P value means that high random overlap with background regions was observed in a number of performed permutations. This information has now been added to the new Statistical Tests section (page 26).

5. Page 9, line 280 onwards: “Similarly, the *ESR1* gene locus exemplifies a hypermethylated DMR, which is associated with loss of local interactions and located at an ER-bound enhancer region in TAMRs and ER+ endocrine resistant clinical tumour samples (n=5) (Fig. 4D). The implication of this statement is that this region would not be hypermethylated in ER+ endocrine sensitive clinical tumour samples – have the authors checked whether this is the case?”

Response:

The methylation patterns in n = 4 primary ER+ tumours have now been added to the figures (**new Figure 4c and d**). The primary tumours are considered endocrine sensitive, because the metastatic tumours were obtained a long time after the treatment start, suggesting initial response to endocrine treatment. Overall, we see a lower methylation level at the DMRs in the primary tumours, but there is a large variability in methylation patterns at these regions between patients. We have additionally provided figures showing percentage of DNA methylation at each of the CpG sites within the DMRs included in the representative examples (**new Supplementary Figure 4c and d**).

Reviewer #2 (Remarks to the Author):

The aim of the manuscript is to address the role of the 3D-chromatin structure in endocrine sensitive and endocrine resistant breast cancer cell lines with further consideration of epigenetic alterations. The manuscript is well written and the scientific question on identifying how ER+ endocrine resistance affects the 3D architecture is interesting and relevant. The approach of combining data sets of not only different samples, but different kind of data reaching from Hi-C, to RNA-seq, and ChIP-seq (and more) offers a great opportunity for in-depth data analysis.

However, I am missing the results that clearly show that the 3D architecture is the key mechanism. I would have expected a suggestion of specific interactions or TADs in combination with relevant genes characterizing the resistance of the two cell lines more specifically.

Response:

In this paper, we characterized changes in 3D structure and their relationship with the epigenome, genome and transcriptome at three main levels of chromatin organisation: chromatin interactions, TADs and A and B compartments. Our findings show that an alteration in 3D genome organisation is associated with development of endocrine resistance. Indeed, we provide examples of some of the most important endocrine resistance genes (e.g. *GREB1* and *ESR1*) whereby differential chromatin interactions create new potential enhancer-promoter interactions associated with a change in gene expression - in addition we identify novel candidate genes that also show differential 3D interactions and gene expression changes. As per Reviewer suggestions, we have

now extended most of our findings to look at associations with gene expression, which allowed us to identify some of the putative target genes for endocrine resistance-associated SNVs. Together, this further supports our findings that changes in 3D architecture are a key mechanism of endocrine resistance and provide mechanistic insights into epigenome alterations and gene de-regulation in endocrine resistant cells.

Moreover, it would be of interest to add an experiment or at least suggest it in the discussion, where important interactions or TADs are mutated (either in the TAMR and FASR cell lines to correct them, or in the MCF7 cell line to introduce them) to see how this influences the regulation of the corresponding genes in the context of endocrine resistant breast cancer.

Another (future) possibility would be also to target the genes affected by interaction/TAD changes using genome editing.

Response:

Although beyond the scope of this work, we agree that it will be interesting in future work to use genome editing approaches, and we have now added this to the discussion. “Future experiments using genome editing to disrupt, delete or introduce TAD boundary elements or anchors of chromatin interactions will establish how identified 3D genome alterations influence the regulation of the corresponding genes in the context of endocrine resistant breast cancer.” (page 18).

Major comments:

1. Three cell lines were used, whereby TAMR and FASR were established from the ER+ MCF7 cell line. The TAMR (tamoxifen resistant) and FASR (fulvestrant resistant) cell lines should only be characterized by the difference characterized by their drug resistance. Thus, it would be helpful to show the RNA-seq data and provide a confirmation that the genes differentially expressed between MCF7 and TAMR, and MCF7 and FASR are associated with the corresponding genes characterizing them.

Response:

The TPM data table from RNA-seq was provided in the GEO submission GSE118713. We have now performed GSEA analyses of the RNA-seq data and significantly enriched MSigDB gene set H (hallmark gene set) results have now been added to **new Supplementary Figure 1c**. As expected, the most significantly altered pathways include estrogen signalling (down in both TAMRs and FASRs) and up-regulation of oncogenic pathways (e.g. EMT, angiogenesis, MYC and ROS).

2. According to Figure 1 three Hi-C libraries were generated for each cell line. However, supplementary table 1 only shows one number per cell line. Thus, it is unclear if the presented number is the mean of each library, or if only one sample was used, or if it is the number of reads combined from all samples. Depending on what is presented, samples should be either further sequenced to achieve a higher number of uniquely mapped reads or a replicate should be added for each cell line. According to the current literature >100 million unique valid pairs are recommended for a replicate (PMID: 22652625).

Response:

We apologise for this lack of clarity. We performed 3 Hi-C replicates per sample. We have now included per replicate information in **new Supplementary Table 1**.

According to the article mentioned by Reviewer, “To look at interactions in *cis* between particular genomic bins at a scale of 100 kb, we recommend obtaining more than 100 million unique valid pairs”, which demonstrates that 100 million valid pairs is the minimum *per sample*. However, these numbers are based on the original, “in ligation” Hi-C protocol, which produced large numbers of artefacts and large proportion of *trans* interactions. Our protocol is based on the “in situ” Hi-C protocol (Rao et al., 2014), which significantly increases the proportion of valid, *cis* interactions in the Hi-C data.

To further support this more recent guidelines including, “The Hitchhiker's Guide to Hi-C Analysis: Practical guidelines” by Kaplan et al., 2015 suggest “adequately complex Hi-C dataset for the human genome with roughly 100 million mapped / valid junction reads, is sufficient to support a 40kb data resolution.”

Therefore, we conclude that our sequencing depth (~ 200 million unique valid pairs per sample) is sufficient for our analyses.

3. It seems that the FASR Hi-C sample has the lowest sequencing depth and the smallest number of unique valid pairs. Based on your conclusion that FASR has less interactions compared to TAMR and MCF7, the question comes up whether this is a true biological consequence due to fulvestrant resistance or if a subset of interaction couldn't be identified due to the lower sequencing depth. Thus, these samples should be sequenced to a more similar depth or the other two samples should be downsampled.

Response:

All performed analyses using diffHic already include normalisation steps that specifically take into account the sequencing depth. diffHic provides new normalization methods to removed trended biases that are abundance-dependent (Lun et al., 2015 <https://doi.org/10.1186/s12859-015-0683-0>). It also implements methods to remove simple scaling biases between libraries and methods to remove genomic biases between interactions and between libraries. Indeed, the FASR cells have more differential interactions as compared to the number of differential interaction identified in TAMR cells (see **Supplementary Table 3**). For data visualisation (both as heatmaps and as arcs), we used Pearson's observed/expected KR normalisation implemented into JuiceBox as described in (Rao et al., 2014). Therefore, differences in sequencing depth do not affect identification of differential interactions or their visualisation.

4. Adding ChIP-seq data to further characterize lost and gained interactions definitely brings in an additional layer of information. Thus, I would recommend to additionally show ChIP-seq profiles (rather than a bar plot as used in Figure 1C).

Response:

We have now provided average binding profiles for ChIP-seq (H3K4me1, H3K27ac) at gained and lost interactions in TAMR and FASR cells (**new Supplementary Figure 1c**, ngs.plot; see Methods) in support of chromHMM data presented in Figure 1c.

5. Figure 1D and 1E: Please provide the genes that are associated with lost and gained interactions along with their expression values as a suppl. table (RNA-seq data).

Response:

We have added **new Supplementary Table 4** with RefSeq gene names, log fold change and FDR-values for all genes located at anchors of differential interactions. Table with TPM values for all genes and all samples has been provided in the GEO submission GSE118713.

6. Figure 1F and 1H, along with the according suppl. figures: Please add the expression levels (from RNA-seq data) to the shown genes. How many folds are they increased/decreased based on RNA-seq? Furthermore, it can't be derived from the figure how the shown RNA-seq reads were normalized nor the track height. Please add this information to allow the comparison of the samples. In general, it would be good to show the fold changes and expression values from the RNA-seq data and provide some ChIP-seq tracks to see if epigenetic changes occur in those regions.

Response:

Thankyou for the suggestions.

We have now added bar plots showing fold changes and P value (t-test) for all genes included in the figures and in the manuscript in **new Supplementary Figure 2**.

We have also added track heights for all figures that include RNA-seq data (**Figure 1f and h, Figure 6d** and matching **Supplementary Figures**).

Information about normalisation of RNA-seq tracks has also been added to the Methods sections (page 25). Tracks for visualisation were generated using bedtools v.2.22 genomeCoverageBed to create normalised .bedGraph files and bedGraphToBigWig (USCS utils) to create .bigwig files. ChromHMM combines all the ChIP-seq tracks and indicates a change in epigenetic states.

7. As far as I know whole genome sequencing of cell lines is very tricky and it seems to me difficult to distinguish pre-existing SNV (those which are present in the cell line, but not related to resistance) from gained ones. The software you used for the analysis suggests to my understanding the usage of matched tumor and normal samples. Thus, it would be helpful if you describe your approach more detailed, but also how you identified the SNVs specific to TAMR or FASR and additional characteristics. How many SNVs did you identify? How many were shared, or specific ones? How did you handle SNV identified in repetitive and black regions of the genome? How many of the SNVs you found were known in dbSNP or other databases and in which context were the known ones identified? Finally, also which genes were affected due to SNVs associated with changed interactions.

Response:

Thankyou for the suggestion. In order to address both reviewers concerns (Reviewer 1 question 2), we have now described our approach in more detail, both in the Methods section (page 23), Results (page 6) as well as provided **new Supplementary Table 5**, which includes details of SNVs identified with Mutect2. Although SNV analyses in cell lines are in deed very difficult, Mutect2 is a recommended software for matched tumour/normal analysis and is suggested for cell line comparison analysis by GATK Best Practises. High coverage (>30x) WGS data from MCF7, TAMR and FASR cells was analysed with Mutect2 using GATK pipeline with following stringent parameters to filter-out any known "germline" variants present in dbSNP v.146, COSMIC v79

coding and non-coding variants (parameter settings: --dbsnp dbsnp_146.hg38.vcf --cosmic CosmicCodingMuts.hg38.vcf --cosmic CosmicNonCodingVariants.hg38.vcf). Therefore, we are confident that the obtained SNVs only include putative endocrine resistance-associated somatic variants (or specific to resistant cell line and not present in control cell line) (total = 14652 in TAMR cells and 15381 in FASR cells). Out of these, 2254 were shared (approx. 15%) was shared between the TAMR and FASR genomes. Summary of somatic SNV calls obtained from Mutect2 has now been included in new **Supplementary Table 5**.

In order to address Reviewer's question about which genes are affected due to SNVs associated with changed interactions, we have now identified 33 putative target genes (29 promoter-coding), which promoters overlapped anchors of differential interactions associated with SNVs. Out of these, in total 19 were differentially expressed in either TAMRs or FASRs as compared to MCF7 (**new Supplementary Figure 3e and f and new Supplementary Table 6**). This suggests that these genes may be further prioritized for functional validation in the future. These results have now been added to the manuscript (page 8).

8. You specifically looked at CTCF. How many loop anchors have CTCF binding? Is it the main TF at your loop anchors or may the SNVs also affect other TF binding motifs, that are maybe also relevant in those samples?

Response:

To characterize the potential effect of SNVs on TF binding motifs, we first gathered the coordinates of SNVs associated with endocrine-resistance (14,652 in TAMR and 15,381 in FASR) and used HOMER annotatePeaks.pl to find their location in relation to genes. Both in TAMR and FASR cells, ~60% of resistance-associated SNVs mapped to intergenic regions, ~35% to introns and ~1% to coding exons (182 in TAMR and 139 in FASR) (**new Supplementary Fig. 3a**). Therefore, in order to identify what regulatory elements may be affected by identified SNVs, we looked at enrichment at MCF7 DNase-Seq sites (ENCODE ENCSR000EPJ). We observed a significant enrichment of TAMR SNVs at DNase-Seq peaks compared to genome-wide background, but not in FASR cells (**new Supplementary Fig. 3c**).

To specifically address what other transcription factor binding sites may be affected by SNVs, we performed observed/expected enrichment analyses using MCF7 ReMap 2018 v.1.2 dataset, which consists of 86 high-quality ChIP-seq datasets. Using MCF7 DNase-Seq peaks as background, we observed that SNVs were enriched at multiple TF binding sites in TAMR and FASR cells, including HSF1, NCOA1/2/3, ESR1, CTCF and FOXA1 binding sites in both TAMR and FASR cells (manuscript new Fig. 2B). Additionally, SNVs in TAMR cells were highly enriched at AR and RAD21 binding, while SNVs in FASR cells were enriched at GATA3 (**new Figure 2b**). The total number of binding sites affected by SNVs for each of the transcription factors is provided in (manuscript **new Figure 3b**).

As CTCF is the most enriched transcription factor at anchors of differential interactions (as shown on **Figure 1c**) and SNVs at CTCF binding sites have been previously shown to affect its binding and interactions, we focused our next analysis on this transcription factor. Additionally, we have also looked at SNVs located at lost ER and FOXA1 binding sites, however the number of SNVs at

these sites was very low (e.g. 17 SNVs at lost ER binding sites and 6 at gained ER binding sites), which significantly reduced our power to detect enrichment.

9. The illustration of the missing loop in Figure 2E should be confirmed using replicates, and/or using a more similar sequencing depth.

Response:

To improve the visual interpretation of the results presented in **new Figure 2e**, we have used WashU browser instead of JuiceBox to illustrate the missing loop. Two replicates per cell line are shown and visualised data has been normalised using the Pearson's observed/expected KR normalisation in JuiceBox. Additional examples have been included in **new Supplementary Fig. 3d**.

10. Figure 3C: It is not clear if you had for all ChIP-seq samples replicates, nor how deep they were sequenced. If you used replicates, how did you handle the replicates for applying peak calling?

Response:

Information about peak calling and intersecting peaks between replicates has already been included in the Methods section and has now been expanded to include more details. For ER binding peak calling all analyses were done with diffBind, taking overlapping peaks between two or three replicates (as described in Methods). Detailed information about peak calling for all ChIP-seq datasets included in the paper has been provided in **new Supplementary Table 7**.

11. Figure 3D: Binding profiles of the ChIP-seq samples would add further information.

Response:

We have now provided average binding profiles for ER ChIP-seq binding profiles at gained and lost interactions in TAMR cells (**new Supplementary Figure 4b**, ngs.plot; see Methods) in support of observed/expected enrichment data presented in **Figure 3d**.

12. Figure 4C and 4D: The expression levels and fold changes of those genes should be shown. Besides ER, H3K27ac would be an informative epigenetic mark.

Response:

Thankyou for the suggestions. We have now added bar plots showing fold changes and P-values for all genes included in the figures and in the manuscript (**Supplementary Figure 2**). The H3K27ac mark has been used for chromHMM state identification and relates to "Active Enhancer" state (H3K4me1 and H3K27ac) in **Figures 4c and d**.

13. TADs gained and lost should be further characterized. What type of genes are in those TADs (lost and gained, also considering TAMR and FASR specific vs shared ones)? How is the gene expression changed within those TADs – any trends? Do chromatin marks change within those TADs? What kind of genes are affected and how? Is it possible to derive gene expression characteristics or a specific pattern regarding epigenetic changes?

Response:

We have now performed additional analyses to fully characterize the TADs gained and lost in endocrine resistance, however the obtained results were not conclusive and further functional studies are required in the future. We identified which TADs are altered in resistant cells due to either gain of a new TAD boundary as compared to MCF7 cells or due to loss of TAD boundary as compared in MCF7 cells. We then intersected these TADs with location of genes in each of the cell types and looked at (1) GO terms enrichment and (2) differential gene expression. For each of the intersected datasets, we obtained a large number of genes, however no significant GO terms were found to be enriched and overall we did not observe a significant change in gene expression (as shown on violin plots on **Rebuttal Figure 3**). We found a large number of genes located within these gained or lost TADs was differentially expressed (~17% of all genes at altered TADs in TAMR and ~18% in FASR), however the number of differentially expressed genes was not above random distribution (Chi-square test). We hypothesise that this inconclusive result is due to large size of identified TADs and we suggest that a more precise CRISPR-Cas9 experiments will be needed to further characterize the role of TAD boundary alterations on gene expression in the future. Indeed the association between TAD structure and gene regulation has been suggested in the past, however more novel results using genome editing techniques and depletion studies shows that TADs are not necessary for correct gene expression patterns to be maintained (Williamson et al., BioRxiv 2019 <https://doi.org/10.1101/609941> as well as CTCF/cohesion depletion studies showing little to no change in gene expression (summaries in Rowley and Corces, 2018 DOI: 10.1038/s41576-018-0060-8).

Rebuttal Figure 3. Differential expression of genes located within altered TADs in TAMR and FASR cells

14. Figures 5 D and E: The pattern of the Hi-C data looks similar. Thus, the usage of Hi-C replicates and/or adjustment of the sequencing depth would give the possibility to further confirm the current results.

Response:

We have now provided per replicate data for results presented in **Figure 5** and this has been added to the **Supplementary Figure 5h-i** in support of main results. All replicates show a visible change in TAD structure, with minor variability between the replicates. Presented data has been normalised for using JuiceBox (KR Pearson's observed/expected scores) and this information has now been added to the Methods section (page 20).

15. Figures 5 D and E: Please add the track height for the ChIP-seq data as well as how they were normalized and an independent control locus.

Response:

Track heights have been added to **Figures 5d and e**. All ChIP-seq datasets were analysed as described in Methods and normalised to input ChIP-seq. A stable CTCF binding site at maintained

TAD boundary in MCF7, TAMR and FASR cells has been marked with an arrow to represent a control locus for ChIP-seq data (**new Figure 5d-e**).

16. How did you take the TAD boundaries identified in Figure 5 into account for identifying the active and inactive compartments?

17. Where are the compartments that change from active to inactive or vice versa located? Can they be associated with changing TADs or interactions? Or from the other perspective – can changed interactions or TADs be associated with switching compartments? What types of genes are within those switching compartments?

Response:

Overall, only a small number of TAD boundaries overlapped with compartment boundaries in each of the cell types (~19% across cell lines), suggesting that TADs are mainly located totally within compartments. Indeed, ~ 60-70% of TADs identified in each cell type was located entirely within compartment. These results have now been added to the manuscript (page 13): “Additionally, TADs identified in each of the cell types were mainly located within the compartment regions (68.21% in MCF7, 68% in TAMR and 60.65% in FASR) with most of the compartment boundaries overlapping a TAD boundary (58.9% in MCF7, 57.8% in TAMR and 62.4% in FASR)”.

Only a very small number (0-10) of lost or gained TAD boundaries in any of the cell types was located at regions that changed its compartment status between A and B and no genome-wide association between compartment switching and altered TAD boundaries was observed. Differential interactions frequently cross TAD boundaries (**Supplementary Fig. 5d**), but are not enriched genome-wide at lost or gained TAD boundaries. In FASRs, lost differential interactions were more often located at regions that switched from A to B compartment status (53 lost DIs) than at regions that switched from B to A (5 lost DIs). Similarly, gained DIs were more often present at compartments that switched from B to A (11 gained DIs) than compartments that switched from A to B (2 gained DIs). This provides some evidence that differential interactions are involved in compartment switching. However, due to small number of overlapping features these new results have not been included in the manuscript as we do not think that they provide informative results.

18. Are there switched compartments which are specific to TAMR or FASR, or are they changed in both? Are specific genes affected?

Response:

We have now performed GO terms analysis for genes located at compartments that switch, but no significant GO terms were identified. This may not be surprising due to the megabase-scale size of compartments containing a large number of ‘passenger’ genes.

More general comments:

19. Figure 1H seems to be missing in the figure legend

Response:

Apologies, A legend has been now added.

20. In those figures where chromatin changes are shown in bar plots, binding profiles would add further information.

Response:

We have now provided average binding profiles for ChIP-seq binding at gained and lost interactions in TAMR and FASR cells (ngs.plot; see Methods page 22) in support of observed/expected enrichment data in the **Supplementary Figure 1c and 3b**.

21. Please add all track heights to the figures to allow a better comparison and describe the normalization used.

Response:

We have added track heights for all figures (Fig. 1f-h, Fig. 2e, Fig. 4c-d, Fig. 5d-e, Fig. 6 d-g and Supplementary Fig. 2 c-h, Supp. Fig. 3d, Supp. Fig. 4e-f and Supp. Fig. 6f-g) and added information about normalisation used to produce bigwig files in the Methods section (page 22) as well as detailed sequencing, mapping and peak calling information in **Supplementary Table 7**.

22. Please add control loci for ChIP-seq data, as they help to see if the reduction in binding is specific to the shown locus.

Response:

A stable CTCF binding site at maintained TAD boundary in MCF7, TAMR and FASR cells has been marked with an arrow to represent a control locus for ChIP-seq data (**new Figure 5d-e**).

23. Besides showing the RNA-seq reads, the RNA-seq expression data along with fold changes should be shown as tables and maybe mentioned in the text.

Response:

We have provided bar plots with TPM values and FDR for all shown genes (**Supplementary Figure 2a-h**). TPM table for all genes from RNA-seq data has been provided in the GEO submission GSE118713.

24. How deep were the other data sequenced? How many replicates were used? How were replicates handled (like peak calling for ChIP-seq; identified interactions; ...)?

Response:

We provided summary information in the specific Methods section on ChIP-seq (page 22) and Hi-C data (page 20). We have also have added a new **Supplementary Table 8** to include additional information on total number of reads, mapping and peak calling for ChIP-seq datasets generated in the study and public datasets. Additionally, detailed per replicate information for generated Hi-C data has been provided in **Supplementary Table 1**.

25. Please also add the used post-hoc tests for statistical tests where multiple comparisons were done.

Response:

Information about all statistical methods has now been added to the Methods section under subsection “Statistical tests”.

26. In line 519: the references seem to be missing.

Response:

Apologies, the missing references have been now provided.

Reviewers' comments:

Reviewer #1 (Remarks to the Author):

The reviewers have addressed several of my concerns, specifically they have generated Hi-C libraries in parental MCF7 cells grown for 0 months, 3 months and 6 months and shown using PCA plots that all of these libraries cluster with the parental MCF7 not the TAMR and FASR cells. They have also made changes to several figures to improve the clarity.

However, I still have some concerns regarding the SNV section of this analysis. If they have excluded SNPs in dbSNP v146 as a filter, then rs201722399 shouldn't be called as an SNV. As I said, I am not an expert in this area but there is some inconsistency that they haven't sorted out – have they used "all SNPs" or "common SNPs" for their filtering? If common SNPs, then they will still have SNPs with $MAF < 1\%$ in their analysis.

The other problem is that they report the sequence flanking rs201722399 as:

CTGTGCCATCTTGG*GACTA

With the G* as their polymorphic variant.

According to UCSC and Ensemble the sequence flanking rs201722399 is:

GAAAAGGGTAAGCA*GAACA

With the A* as the reference allele of their polymorphic variant.

ie the sequence they report bears no resemblance to the sequence flanking rs201722399. I have also tried aligning the opposite strand and that is no better. Maybe I am missing something but I don't think their reported sequence maps to chr17:18,694,656.

I also have concerns with respect to the response to my point 5.

Page 11, line 338 onwards now says

"Hypermethylation at the ER-enhancer regions where chromatin interactions are lost in the TAMRs relative to MCF7 cells can be observed in a zoomed-in view of each of the identified DMRs. A gain of methylation is also commonly found at these loci in ER+ endocrine resistant clinical tumour samples (n=5) as compared to primary tumours (n=4) (Fig. 4c and Supplementary Fig. 4b)."

I think in fact this should be Fig 4c and Supplementary Fig 4c (for NCOR2). If I have understood correctly, these methylation plots should demonstrate that the green dots are, on average, lower than the red dots if what they claim is true. I can believe that for ESR1 (Sup Fig 4d) – but not for any of the others (NCOR2 in Sup Fig 4c and MSI2 Sup Fig 4g).

To see if my eye is deceiving me I did a very crude analysis of NCOR2 DMR1 and DMR2. I took the mean of the methylation measurements (from the supplementary data) for each primary (n=4) and each met (n=5) and then looked at the mean of these means.

DMR1, primary mean= 21.37, sd=8.26; met mean=18.3, sd=17.5

DMR2, primary mean=40.0, sd=17.3; met mean=32.5, sd=28.2

If anything, the primaries seem to have higher average methylation. So, what is the evidence that the NCOR2, ESR1 and MSI2 DMRs are hypermethylated in the endocrine resistant samples compared to the primary samples? How have they analysed these data, what is the mean (?) median (?) methylation for these specific DMRs that they calculate and what is the P-value demonstrating a difference?

Reviewer #2 (Remarks to the Author):

The authors answered all my question to my complete satisfaction. I would only like to suggest some smaller additions.

1) The HiC data of the MCF7 cell lines (T0, T3, T6) are now from a different experiment, which does not necessarily mean that they are the same, as you might have a batch effect. Where the data corrected for bath effects? The PCA plots in suppl. figure 1a show that the three MCF7 cells cluster closer together than the MCF7 cells at the different time points and the public available one. What are the driving genes for this difference between the MCF7 cells, but also between the four categories of MCF7 cells and the TAMR and FASR cell lines. Do similar gene expression patterns appear?

2) Line 118: Sentence is twice

3) Figure 1f: the RefSeq track should be shown like in figure 1h, where genes are not overlapping. Right now, it is difficult to distinguish Mir4429 from GREB1.

4) Supplementary Figure 1b: There are the two different coverage plots for lost and gained interactions. Please describe the two plots – like replicate 1 and 2 or enhancer and promotor regions.

5) Supplementary Figure 1: figure legends for panel c is missing; panel d and e need to be adapted to the new figure and f is missing again.

6) You claim that the anchors are enriched/reduced for enhancer and promotor marks. H3K4me1 is rather an enhancer mark than a promotor mark. Thus, H3K4me3 should be added for the promotor regions. How does H3K4me3 looks at those genes?

7) Suppl. Figure 5h and i: It would help to add the same information as in figure 5d and e here, to make the lost and gained TADs more visible (also the corresponding CTCF binding sites)

8) Suppl. figure 6: add the track height

9) Statistical analysis: thank you for adding all the statistical information. However, I am still missing a section for describing the multiple comparisons (applied test, parametric or non-parametric, ...) as applied in figure 6c. Please add this information to the materials and methods.

10) The supplementary tables are difficult to distinguish, as the table number is not visible (maybe also due to the upload). I guess, in general, it helps if you add a short header to the tables, as you did in the Word file describing the suppl. tables and figures.

We thank the reviewers for their letter advising that our manuscript was significantly improved and that we have addressed most of their concerns. We have now provided additional evidence and analysis to address the remaining concerns from Reviewer 1. Additionally, we have updated Figures and Tables to include the small additions suggested by Reviewer 2.

We have marked changes in the manuscript in green.

Please find a summary of our responses below:

Reviewer #1 (Remarks to the Author):

The reviewers have addressed several of my concerns, specifically they have generated Hi-C libraries in parental MCF7 cells grown for 0 months, 3 months and 6 months and shown using PCA plots that the all of these libraries cluster with the parental MCF7 not the TAMR and FASR cells.

They have also made changes to several figures to improve the clarity.

However, I still have some concerns regarding the SNV section of this analysis. If they have excluded SNPs in dbSNP v146 as a filter, then rs201722399 shouldn't be called as an SNV. As I said, I am not an expert in this area but there is some inconsistency that they haven't sorted out – have they used “all SNPs” or “common SNPs” for their filtering? If common SNPs, then they will still have SNPs with MAF<1% in their analysis.

Response:

We think that part of the confusion arises as we are comparing the MCF7 breast cancer cell line with a derived resistant cell line and used the MuTect2 to identify SNVs that were present only in the resistant cell line. MuTect2 is primarily used to identify somatic SNVs between matched normal and tumour samples. However in our case we used MuTect 2 to identify SNVs only present in the endocrine resistant cell lines by designation of the “Normal” cell as MCF7 and the “Tumour” as the FASR/TAMR cells.

dbSNP resource includes all kinds of genetic variations (including germline, somatic and of unknown origin) (as explained:

<https://www.ncbi.nlm.nih.gov/books/NBK44447/>). Thus, SNVs cannot be classed as either germline or somatic only based on their presence or absence in dbSNP database.

Specifically rs201722399 variant in dbSNP **was not filtered** out as a germline SNV because it was classified as “unspecified category” in dbSNP:

chr17:18694656 (rs201722399): SAO=0, OTHERKG

dbSNP Classification: SAO = "Variant Allele Origin: 0 - unspecified, 1 - Germline, 2 - Somatic, 3 - Both", OTHERKG = "non-1000 Genome submission (not present at 1000 Genomes Project"

In MuTect2 analysis **all variant sites** present in the dbSNP v.146 resource are used to aid filtering of known germline variants, leaving somatic and unspecified variants. MuTect2 pipeline does not automatically filter-out dbSNP variants and only uses the dbSNP overlap to require more evidence through a more strict LOD threshold (--dbsnp_normal_lod 5.5) (as explained: https://software.broadinstitute.org/gatk/documentation/tooldocs/3.8-0/org_broadinstitute_gatk_tools_walkers_cancer_m2_MuTect2.php#--dbsnp_normal_lod).

In our analysis using the MuTect2 pipeline the SNV (rs201722399, A/G*) was called a “somatic” variant in the resistant cell line (FASRs) as it met all the requirements. This includes (1) A to G substitution is not present in the “normal” MCF7 cells (27/27 reads are “A”) but is present in the “tumour” resistant cells (33/40 reads are “A”, 1 is “C” and 6 are “G”) and (2) that the LOD threshold has been met in both the “normal” sample (NLOD = 6.62) and “tumour” sample (TLOD = 11.35). Additionally, (3) the MAF recorded for this variant in gnomAD is < 1% cut-off (Hiltemann et al., 2015 Genome Res; doi: 10.1101/gr.183053.114) (MAF = 0.003114).

We hope that together this explains that we used **all dbSNP short human variations** formatted in VCF for filtering and why rs201722399 is not filtered out using the MuTect 2 pipeline.

We have now explained further in the methods the filtering process to identify endocrine-associated SNVs on page 24 “SNVs that have been previously identified in dbSNP v.146 were included if they fulfilled all the requirements of MuTect2 to be called somatic⁵⁷, including that they were not previously reported as germline (dbSNP SAO flag “germline”)”.

The other problem is that they report the sequence flanking rs201722399 as:

CTGTGCCATCTTGG*GACTA

With the G* as their polymorphic variant.

According to UCSC and Ensemble the sequence flanking rs201722399 is:

GAAAAGGGTAAGCA*GAACA

With the A* as the reference allele of their polymorphic variant.

ie the sequence they report bears no resemblance to the sequence flanking rs201722399. I have also tried aligning the opposite strand and that is no better. Maybe I am missing something but I don't think their reported sequence maps to chr17:18,694,656.

Apologies for the confusion regarding the sequence we provided for the rs201722399 variant. The sequence we included by mistake is from the adjacent SNV within the highlighted region. We have now provided the correct flanking sequence for the rs201722399 variant in **new Figure 2d and e**. Additionally, we included a screen-

shot of interactions at the other CTCF-altering SNVs on chromosomes 4 and 5 as well as provided zoomed-in views of all SNVs at CTCF binding sites for all of the identified candidate SNVs (**new Supplementary Fig. 3d and e**).

2. I also have concerns with respect to the response to my point 5.

Page 11, line 338 onwards now says

“Hypermethylation at the ER-enhancer regions where chromatin interactions are lost in the TAMRs relative to MCF7 cells can be observed in a zoomed-in view of each of the identified DMRs. A gain of methylation is also commonly found at these loci in ER+ endocrine resistant clinical tumour samples (n=5) as compared to primary tumours (n=4) (Fig. 4c and Supplementary Fig. 4b).”

I think in fact this should be Fig 4c and Supplementary Fig 4c (for NCOR2). If I have understood correctly, these methylation plots should demonstrate that the green dots are, on average, lower than the red dots if what they claim is true. I can believe that for ESR1 (Sup Fig 4d) – but not for any of the others (NCOR2 in Sup Fig 4c and MSI2 Sup Fig 4g).

To see if my eye is deceiving me I did a very crude analysis of NCOR2 DMR1 and DMR2. I took the mean of the methylation measurements (from the supplementary data) for each primary (n=4) and each met (n=5) and then looked at the mean of these means.

DMR1, primary mean= 21.37, sd=8.26; met mean=18.3, sd=17.5

DMR2, primary mean=40.0, sd=17.3; met mean=32.5, sd=28.2

If anything, the primaries seem to have higher average methylation. So, what is the evidence that the NCOR2, ESR1 and MSI2 DMRs are hypermethylated in the endocrine resistant samples compared to the primary samples? How have they analysed these data, what is the mean (?) median (?) methylation for these specific DMRs that they calculate and what is the P-value demonstrating a difference?

Response:

We report an overall higher methylation trend at the DMRs in the metastatic tumours based on visualisation of candidate DMRs previously described (Stone et al., 2015 doi: 10.1038/ncomms8758) but there clearly is a large variability in methylation patterns at these regions between patients. To further clarify the point about patient variability we have now provided per CpG site methylation values for each of the matched pairs of primary (i.e. endocrine sensitive) and metastatic (i.e. endocrine resistant) tumour samples in the **new Supplementary Fig. 4d-e and h**. For example the case of NCOR2 example in DMR1 only 1 or 2 patients show a trend of increased methylation at these sites, but for DMR 3, most sites have increased methylation between primary and metastatic matched samples (**new Supplementary Fig. 4d**). However, due to limited number of samples and high variability between patient samples, no statistically significant differences are observed.

In some cases where high methylation values are already observed at these DMRs in primary tumour samples, it may suggest intrinsic or *de novo* resistance is already present in the primary tumour.

We have now modified the manuscript on Page 11 to further clarify the methylation patterns in the clinical samples: “A gain of methylation is also observed at some of these loci in ER+ endocrine resistant clinical tumour samples (n=4) as compared to matched primary tumours from the same patients (n=4) (Fig. 4c and Supplementary Fig. 4d). Similarly, the *ESR1* (Fig. 4d and Supplementary Fig. 4e) and *MSI2* (Supplementary Fig. 4f-h) gene loci exemplify a hypermethylated DMR, which is associated with loss of local interactions and located at an ER-bound enhancer region in TAMRs and ER+ endocrine resistant clinical tumour samples as compared to their matched primary tumours. However, there is large variability in DNA methylation observed between patient samples at each of the CpG sites within the shown DMRs. In some cases where there is hypermethylation in the primary tumour, may suggest intrinsic or *de novo* methylation at these sites, which impacts the susceptibility to acquire endocrine resistance^{6, 36}”.

We have also updated the Source Data file to reflect the pairing between the primary and metastatic samples.

Additionally, we have now analysed ER binding changes in clinical samples from patients who responded to endocrine therapy (Responders) and from patients who did not respond to endocrine treatment (Non-responders and Metastatic) (GEO: GSE32222 Ross-Innes et al., 2012 Nature) and we compared methylation levels in primary and metastatic samples at these sites. The new results in **new Supplementary Fig. 4c** further support our findings that DNA methylation changes in endocrine resistance are associated with altered ER binding. Page 11 now reads: “We then investigated if the association between DNA methylation and ER binding is maintained in clinical samples. Using ER ChIP-seq binding in primary breast cancers from patients with different clinical outcomes (9 “Responders” vs. 9 “Non-responders” and 3 “Metastatic tumours”), we identified ER binding sites, which were lost in non-responders (n = 14,553) and gained in non-responders (n = 1662). We then examined the DNA methylation profiles from primary (i.e. endocrine sensitive) and metastatic (i.e. endocrine-resistant) breast tumours around lost and gained ER binding sites that were located at the ERE motif. We observed a significant gain of methylation in metastatic samples at ER binding sites lost in non-responders (Mann-Whitney P value = 0.0106) and an overall loss of methylation at gained ER binding sites (not significant) (Supplementary Fig. 4c).”

Reviewer #2 (Remarks to the Author):

The authors answered all my question to my complete satisfaction. I would only like to suggest some smaller additions.

1) The HiC data of the MCF7 cell lines (T0, T3, T6) are now from a different experiment, which does not necessarily mean that they are the same, as you might have a batch effect. Where the data corrected for bath effects?

Response:

In order to account for potential batch effects between the “old” and “new” MCF7 Hi-C data, we processed the data using diffHic between-sample normalisation steps, which is based on the idea of loess normalization from gene expression microarrays (Yang, Dudoit, et al., 2002). In this method, the estimated fold-changes between conditions are modelled using a loess smoother as a function of average contact strength (Lun and Smyth, 2015). Using the estimated model, the data are corrected so there is no effect of the covariate on the fold-change.

The PCA plots in suppl. figure 1a show that the three MCF7 cells cluster closer together than the MCF7 cells at the different time points and the public available one. What are the driving genes for this difference between the MCF7 cells, but also between the four categories of MCF7 cells and the TAMR and FASR cell lines. Do similar gene expression patterns appear?

We do not have gene expression data for all of the different MCF7 cells (n = 10) used in the PCA plots and therefore we are unable to check if there are different gene expression patterns. Importantly in our model, the same parental MCF7 cells were used to generate the resistant derivatives (TAMR and FASR) and we used expression data for these cells in our paper to show differentially expressed genes were enriched for pathways known to be associated with endocrine resistance (e.g. Estrogen response) and cancer (e.g. EMT, angiogenesis) (Supplementary Fig. 1c).

2) Line 118: Sentence is twice

Response:

We have now modified both sentences. This paragraph now reads: “Interestingly, all differential interactions were significantly enriched for enhancer and promoters, as well as CTCF sites, regardless of the TAMR or FASR treatment regime (Fig. 1c). However, gained chromatin interactions in TAMR and FASR cells showed higher enrichment of active enhancer marks (H3K4me1 and H3K27ac), compared to lost interactions (Supplementary Fig. 1b). Similarly there was increased enrichment of the active promoter mark H3K4me3 at gained interactions in in TAMR and FASR cells relative to MCF7 cells (Supplementary Fig. 1b).”

3) Figure 1f: the RefSeq track should be shown like in figure 1h, where genes are not overlapping. Right now, it is difficult to distinguish Mir4429 from GREB1.

Response:

The gene transcripts track has now been expanded and miRNA-4429 can now be distinguished from GREB1 (**new Fig. 1f**).

4) Supplementary Figure 1b: There are the two different coverage plots for lost and gained interactions. Please describe the two plots – like replicate 1 and 2 or enhancer and promotor regions.

Response:

The plots show average profile of H3K4me1 (left plot) and H3K27ac (right plot) for all gained (left side) and lost (right side) interactions. We have now separated plots for enhancer (H3K4me1 and H3K27ac) and promoter (H3K4me3) marks, added titles to the plots and described them in the figure legend (**new Supplementary Fig. 1b**).

5) Supplementary Figure 1: figure legends for panel c is missing; panel d and e need to be adapted to the new figure and f is missing again.

Response:

Thank you. We have now updated the legend for Supplementary Figure 1 d and e and added the legend to panel c and f.

6) You claim that the anchors are enriched/reduced for enhancer and promotor marks. H3K4me1 is rather an enhancer mark than a promotor mark. Thus, H3K4me3 should be added for the promotor regions. How does H3K4me3 looks at those genes?

Response:

We have now added new average profile plots for H3K4me3 at all gained and lost interactions in TAMR and FASR cell lines for putative promoter regions. We have also now separated the enhancer mark plots (H3K4me1 and H3K27ac) and promoter mark (H3K4me3) plots (**new Supplementary Fig. 1b**).

7) Suppl. Figure 5h and i: It would help to add the same information as in figure 5d and e here, to make the lost and gained TADs more visible (also the corresponding CTCF binding sites)

Response:

We have now modified the **new Supplementary Fig. 5h** and **i** as suggested.

8) Suppl. figure 6: add the track height

Response:

Thank you. We have now added track heights for screenshots in **new Supplementary Fig. 6e-g**.

9) Statistical analysis: thank you for adding all the statistical information. However, I am still missing a section for describing the multiple comparisons (applied test, parametric or non-parametric, ...) as applied in figure 6c. Please add this information to the materials and methods.

Response:

In Figure 6c the data was compared independently between the groups (i.e. Stable vs. A to B and Stable vs. B to A) and therefore a non-parametric Wilcoxon rank-sum test was applied. This information is already included in the Methods section: “The Mann–Whitney–Wilcoxon tests were used for 2-group non-parametric comparisons.”

10) The supplementary tables are difficult to distinguish, as the table number is not visible (maybe also due to the upload). I guess, in general, it helps if you add a short header to the tables, as you did in the Word file describing the suppl. tables and figures.

Response:

We have now added headers to Supplementary Tables when it was possible (**Supplementary Table 4,6,7 and 8**). We have not added these to .tsv files (i.e. Supplementary Table 2 and Supplementary Table 3) as it will make it difficult to work with these files in programs like R etc.

Reviewers' comments:

Reviewer #1 (Remarks to the Author):

The authors have attempted to respond to my comments.

I am not qualified to comment further on their SNV analysis.

I have remaining concerns regarding the comparison methylation status between clinical samples. I don't think it's appropriate to report differences based on visualisation that are not statistically significant but this is an editorial decision.

Reviewer #3 (Remarks to the Author):

The core of this paper is the Hi-C measurements on MCF7 cells together with MCF7 cells grown to be resistant to two different drugs.

I have found one critical technical flaw (which can be corrected, but IMO is critical to correct prior to publication). In my opinion all technical issues raised by the prior reviewers have been addressed, with the exception of the critical flaw which is related to one of these issues. To me, acceptance or rejection of this manuscript resides solely on an editorial decision of impact. I will comment on these things below.

Re. Impact

This paper focuses the bulk of the work (and all Hi-C measurements) on cell lines, for technical and monetary reasons. In this it follows the Hi-C field; the majority of published experiments in humans are done using cell lines. For cancer applications, the choice of cell lines can always be debated, but in this manuscript there is a nice causal treatment intervention giving the observations a strong causal interpretation; that they are caused by the selection for resistance. This is a strength of the paper.

Regarding causality, there is no data in the paper which sheds light on the causal ordering of molecular changes - does differential interactions lead to differential expression or methylation or is it the other way around? In spite of this, the paper several places uses language which makes it appear that a causal ordering has been established. I respect the authors right to speculate and hypothesize on this, but the current writing does not make it clear that it is indeed speculation and hypothesizing. In light of this, the language could be substantially improved.

A major part of the paper is relating changes in interactions to changes in expression/methylation/somatic genetic changes, and this is primarily done via enrichment analyses. The pattern is somewhat disappointing, but probably expected. In my interpretation we see other molecular changes at anchors of differential interactions, but there are many differential interactions that do not lead to changes in expression/methylation etc. and there are many changes in expression/methylation etc which does not lead to differential interactions. I would not say this is the authors interpretation, but to me that is pretty clearly what the data suggests. In support of this, my read of the various enrichment estimates is that they are significant, but that the enrichment estimate is quite modest for assays done on the same sample. An enrichment of 2-3 (here I assume the authors mean odds-ratio) is in my opinion modest, not as the authors write "high enrichment".

The Hi-C data does not actually lead to any real biological insight in my reading, apart from (1) resistance is associated with some changes in interactions and (2) that differential interactions are

sometimes related to other molecular changes.

All of this decreases my _excitement_ about the results, but on the other hand this is a well done experiment and it has a good casual interpretation - its just not the case that 3D interactions suddenly makes everything make sense. The editorial decision is therefore whether solid incremental science without massive overhype has a place in Nat Comms. Personally, I would hope so.

The critical flaw of the manuscript is the PCA figure 1A which was added (?) or at least is the answer to previous reviewers question re. cell culture and batch effects. The figure is beautiful, but reading the methods section carefully, I was extremely surprised to see that the PCA was done on differential interactions. This is wrong and implies that the figure cannot be used to support the question asked, because it becomes an example of supervised PCA. To be correct, the analysis needs to be unsupervised. In differential expression one would usually use the top X% most variable genes (or perhaps all expressed genes) to make this point and not (as is currently done) the differentially expressed genes. Now, it is very likely that the new PCA plot will look much, much less conclusive and perhaps no longer strongly support that batch / culture has minimal impact on the results. I want to state up front that most current HiC papers ignore this completely, so any investigation of this is in fact an improvement on most existing papers.

The comments from Reviewer 1 about mistakes / errors in flanking sequences appear to have been resolved. I commend the reviewer for finding these mistakes, but the authors have said it was a simple error and have corrected it.

OTHER QUESTIONS WHICH CAN EASILY BE ADDRESSED BY WRITING

1) I have a question however regarding the mutation analysis. I understand the broad question of how you're using mutect2 to infer variants related to resistance by comparing MCF7 and the other cell lines. What I don't understand here is why the focus on whether the variant is known to occur in the gremline. You have selection mechanism going on, and why could resistance associated mutations not occur in the gremline. This is not about causing cancer but about whether the mutation is associated with resistance. I find this part of the paper puzzling and I don't fully understand why. I acknowledge that the methods section specify exactly how you did it, but without reading the mutect2 manual and paper, the reader cannot really understand the purpose is. The response to the reviewer I have seen, is more helpful but is not reflected in the manuscript.

2) In general, when doing enrichment analysis I would like to see the numbers of actual overlap and not just the enrichment.

3) What scale is "enrichment" on? I think it is odds-ratios but I don't see it described, only indirectly by saying you're using GAT.

4) If I understand the manuscript correctly, you don't actually do differential expression testing genomewide, only for the genes involved in interactions. That means we don't know how many genes were DE without being involved in an interaction. Am I completely misunderstanding this? I understand the multiple testing burden is different, but I would still like to see a genome wide DE test. I suspect many genes are DE without being involved in interactions (which I am fine with, I just want the number).

Reviewer #1: *I have remaining concerns regarding the comparison methylation status between clinical samples. I don't think it's appropriate to report differences based on visualisation that are not statistically significant but this is an editorial decision.*

Response:

Initially Reviewer #1 asked in their comment 5 “*Similarly, the ESR1 gene locus exemplifies a hypermethylated DMR, which is associated with loss of local interactions and located at an ER-bound enhancer region in TAMRs and ER+ endocrine resistant clinical tumour samples (n=5) (Fig. 4D).* The implication of this statement is that this region would not be hypermethylated in ER+ endocrine sensitive clinical tumour samples – have the authors checked whether this is the case.

Indeed when we finally identified some primary/matched metastatic tumour samples to address this question we found that in the ESR1 enhancer region there **was significantly less methylation** in the primary sample relative to the metastatic samples (Wilcoxon paired test $P = 0.02$) (new Supp. Fig. 4e). This directly addressed Reviewer #1 question - however we then went on to show that there was a high degree of variability in the other regions. We presented these as examples to show similar trends not statistically significant differences.

We initially chose not to add matched primary data to our original submission as 1) we only had a limited number of matched primary and metastatic samples (and no public data available) and therefore obtaining significance in the data between primary and resistance is difficult - especially in consideration that 2) the presence of potential intrinsic resistance in primary tumours and 3) high variability between patients.

We have sourced 3 more matched pairs and with added sample numbers ($n=7$) and the meta-analysis of the data we now show 3 regions within each of the 3 genes (ESR1 ($P<0.001$), NCOR2 ($P<0.001$), MSI2 ($P=0.0312$); Wilcoxon rank test) reach significance – we can show this in a revised manuscript.

Reviewer #1: *I am not qualified to comment further on their SNV analysis.*

Response:

Further to our response to Reviewer's #1, we note that Jung et al., Nature Biotech, 2013 “*Systematic investigation of cancer-associated somatic point mutations in SNP databases*” paper directly addresses the issue with filtering identified SNVs against public SNP databases.

This paper showed that 50% of cancer-associated somatic mutations found in COSMIC (Catalogue of **Somatic** Mutations in Cancer) were found in dbSNP, demonstrating the danger of filtering using dbSNP and stressing the need for better filtering strategies.

The authors suggest filtering out only those common SNPs with a $MAF > 1\%$ as one way of ameliorating the problem. This is exactly the approach we have used in the SNV analysis in our study.

We have marked changes in the manuscript in red.

Please find a summary of our responses below in order of priority:

Reviewer #3 (Remarks to the Author):

The core of this paper is the Hi-C measurements on MCF7 cells together with MCF7 cells grown to be resistant to two different drugs.

I have found one critical technical flaw (which can be corrected, but IMO is critical to correct prior to publication). In my opinion **all technical issues raised by the prior reviewers have been addressed**, with the exception of the critical flaw which is related to one of these issues. To me, acceptance or rejection of this manuscript resides solely on an editorial decision of impact. I will comment on these things below.

The critical flaw of the manuscript is the PCA figure 1A which was added (?) or at least is the answer to previous reviewers question re. cell culture and batch effects. The figure is beautiful, but reading the methods section carefully, I was extremely surprised to see that the PCA was done on differential interactions. This is wrong and implies that the figure cannot be used to support the question asked, because it becomes an example of supervised PCA. To be correct, the analysis needs to be unsupervised. In differential expression one would usually use the top X% most variable genes (or perhaps all expressed genes) to make this point and not (as is currently done) the differentially expressed genes. Now, it is very likely that the new PCA plot will look much, much less conclusive and perhaps no longer strongly support that batch / culture has minimal impact on the results. I want to state up front that most current HiC papers ignore this completely, so any investigation of this is in fact an improvement on most existing papers.

Response:

The reviewer is confused here - the MDS plots were performed on the top most variable 1000 chromatin conformation contacts (i.e. “different interactions”) and not statistically significant differential interactions between cell lines.

Contrary to Reviewer’s 3 interpretation, MDS plots were constructed in an **unsupervised** manner with the plotMDS function applied to all filtered and normalized count values for each bin pair for each library. This is done before testing for significant differential interactions and **not** using significant differential interactions (similar to RNA-seq pipelines). Therefore our analyses were already performed in the exact way requested by Reviewer. We agree this is the correct method to examine the similarities between Hi-C libraries and has been frequently used in other publications using Hi-C data (1-4) as well as RNA-seq. Thus our original conclusions about cell culture/batch effect impact on 3D interactions are substantiated.

We appreciate that the use of the word “differential interaction” in the Methods might have been confusing here and so we have now edited this to say “top 1000 most variable bin pairs” and expanded the section to provide more details “*Multidimensional-scaling plots were constructed using plotMDS function applied to diffHiC processed filtered and normalised counts for each bin pair for each library. The distance between each pair of libraries is computed as the ‘leading log fold change’, defined as the root-mean-square average of the top 1000 most variable bin pairs at 5Mb, 1Mb and 100kb resolution*”. See page 21 Methods.

The comments from Reviewer 1 about mistakes / errors in flanking sequences appear to have been resolved. I commend the reviewer for finding these mistakes, but the authors have said it was a simple error and have corrected it.

Response: Thankyou.

OTHER QUESTIONS WHICH CAN EASILY BE ADDRESSED BY WRITING

1) I have a question however regarding the mutation analysis. I understand the broad question of how you’re using mutect2 to infer variants related to resistance by comparing MCF7 and the other cell lines. What I don’t understand here is why the focus on whether the variant is known to occur in the germline. You have selection mechanism going on, and why could resistance associated mutations not occur in the germline. This is not about causing cancer but about whether the mutation is associated with resistance. I find this part of the paper puzzling and I don’t fully understand why. I acknowledge that the methods section specify exactly how you did it, but without reading the mutect2 manual and paper, the reader cannot really understand the purpose is. The response to the reviewer I have seen, is more helpful but is not reflected in the manuscript.

Response: We agree that resistance associated mutations could potentially occur in the germline. However both Reviewer#1 and Reviewer#2 requested that we exclude any potential germline variants, as they considered them not informative (see Rebuttal 1, Reviewer#1 question 2 and Reviewer#2 question 7). Both Reviewers also expressed their concern that “whole genome sequencing of cell lines is very tricky and it seems difficult to distinguish pre-existing SNV (those which are present in the cell line, but not related to resistance) from gained ones” (Reviewer#1).

Therefore to address their concerns and reduce the risk of detecting false positives, we excluded any SNVs that were marked as “germline risk” in MuTect2 output and focused only on high confidence mutations associated with resistance.

We have now included in the Methods (page 24) the response to the reviewer “*In MuTect2 analysis all variant sites present in the dbSNP v.146 resource and COSMIC v79 coding and non-coding variants are used to aid filtering of known germline variants, leaving somatic and unspecified variants.*”

2) In general, when doing enrichment analysis I would like to see the numbers of actual overlap and not just the enrichment.

Response: The numbers of actual overlapping features are provided on the respective figures (e.g. Fig. 2a and b, Fig. 3d, Fig 4a, Fig. 5c and Fig 6f) as previously requested by Reviewer#2 question 4. For Fig. 1c this data is provided in the Source Data file.

3) What scale is “enrichment” on? I think it is odds-ratios but I don’t see it described, only indirectly by saying you’re using GAT.

Response: Presented enrichment is a fold change from a permutation test performed using GAT (5) and not odds-ratio. To clarify this, we have **now** updated Y-axis labels as well as figure legends for Fig. 1c, 2a-b, 3d, 4a, 5c, 6f and Supp. Fig. 3c and 5g.

4) If I understand the manuscript correctly, you don’t actually do differential expression testing genomewide, only for the genes involved in interactions. That means we don’t know how many genes were DE without being involved in an interaction. Am I completely misunderstanding this? I understand the multiple testing burden is different, but I would still like to see a genome wide DE test. I suspect many genes are DE without being involved in interactions (which I am fine with, I just want the number).

Response: Genome-wide analyses of RNA-seq data were already performed as described in Methods and were provided in Supp. 1c as per request of Reviewer 2 question 1. We have **now** added the total number of differentially expressed genes obtained from differential analysis in edgeR (FDR < 0.05; logFC > 4) in the Supp. Fig. 1c.

Other Comments:

Re. Impact

This paper focuses the bulk of the work (and all Hi-C measurements) on cell lines, for technical and monetary reasons. In this it follows the Hi-C field; the majority of published experiments in humans are done using cell lines. For cancer applications, the choice of cell lines can always be debated, but in this manuscript there is a nice causal treatment intervention giving the observations a strong causal interpretation; that they are *_caused_* by the selection for resistance. **This is a strength of the paper.**

Regarding causality, there is no data in the paper which sheds light on the causal ordering of molecular changes - does differential interactions lead to differential expression or methylation or is it the other way around? In spite of this, the paper several places uses language which makes it appear that a causal ordering has been established. I respect the authors right to speculate and hypothesize on this, but the current writing does not make it clear that it is indeed speculation and hypothesizing. In light of this, the language could be substantially improved.

Response: We have revised the manuscript to ensure that we do not say our experiments *prove* the causal order of changes. We are careful throughout to say that that our data is ‘associated’ with decreased ER binding and atypical interactions and gene expression and that we therefore ‘suggest’ that 3D epigenome remodelling is a key mechanism underlying endocrine resistance in ER+ breast cancer. Additionally, as already requested by Reviewer#1 (comment 2) the following sentence has been

already added to the Discussion “*Future experiments using genome editing to disrupt, delete or introduce TAD boundary elements or anchors of chromatin interactions will establish how identified 3D genome alterations influence the regulation of the corresponding genes in the context of endocrine resistant breast cancer.*” (page 18).

A major part of the paper is relating changes in interactions to changes in expression/methylation/somatic genetic changes, and this is primarily done via enrichment analyses. The pattern is somewhat disappointing, but probably expected. In my interpretation we see other molecular changes at anchors of differential interactions, but there are many differential interactions that do not lead to changes in expression/methylation etc. and there are many changes in expression/methylation etc which does not lead to differential interactions. I would not say this is the authors interpretation, but to me that is pretty clearly what the data suggests. In support of this, my read of the various enrichment estimates is that they are significant, but that the enrichment estimate is quite modest for assays done on the same sample. An enrichment of 2-3 (here I assume the authors mean odds-ratio) is in my opinion modest, not as the authors write “high enrichment”.

Response: We performed genome-wide analysis and association studies between different molecular changes (3D genome, histone marks, TF binding, genetic variants, DNA methylation and gene expression) using a simulation framework for testing the association of genomic intervals (5). The observed enrichment is statistically significant (multiple testing corrected FDR < 1% for most of the enrichments; n = 1000 permutations). As all these datasets encompass the whole genome, we do not expect a one-to-one relationship between these changes. Previous studies performed either genome-wide (1, 3, 6, 7) or using a single-locus (8) have shown that small changes in 3D interactions can have large effect on gene expression and cause large phenotypic changes (8, 9). We have now modified “high enrichment” in the text to “significant enrichment” (see page 13).

The Hi-C data does not actually lead to any real biological insight in my reading, apart from (1) resistance is associated with some changes in interactions and (2) that differential interactions are sometimes related to other molecular changes.

Response: The key biological insights presented in the paper are that (1) resistance is associated with differential interactions and that (2) these differential interactions are associated with other genetic (SNVs) and epigenetic (DNA methylation, histone marks, ER binding) changes.

References cited above:

1. Johanson TM, Lun ATL, Coughlan HD, Tan T, Smyth GK, Nutt SL, et al. Transcription-factor-mediated supervision of global genome architecture maintains B cell identity. *Nat Immunol.* 2018;19(11):1257-64.
2. Jansz N, Keniry A, Trussart M, Bildsoe H, Beck T, Tonks ID, et al. Smchd1 regulates long-range chromatin interactions on the inactive X chromosome and at Hox clusters. *Nature structural & molecular biology.* 2018;25(9):766-+.

3. Greenwald WW, Li H, Benaglio P, Jakubosky D, Matsui H, Schmitt A, et al. Subtle changes in chromatin loop contact propensity are associated with differential gene regulation and expression. *Nature Communications*. 2019;10.
4. D'ippolito AM, McDowell IC, Barrera A, Hong LK, Leichter SM, Bartelt LC, et al. Pre-established Chromatin Interactions Mediate the Genomic Response to Glucocorticoids. *Cell Systems*. 2018;7(2):146-+.
5. Heger A, Webber C, Goodson M, Ponting CP, Lunter G. GAT: a simulation framework for testing the association of genomic intervals. *Bioinformatics*. 2013;29(16):2046-8.
6. Flavahan WA, Drier Y, Johnstone SE, Hemming ML, Tarjan DR, Hegazi E, et al. Altered chromosomal topology drives oncogenic programs in SDH-deficient GIST. *Nature*. 2019.
7. Flavahan WA, Drier Y, Liao BB, Gillespie SM, Venteicher AS, Stemmer-Rachamimov AO, et al. Insulator dysfunction and oncogene activation in IDH mutant gliomas. *Nature*. 2016;529(7584):110-4.
8. Lupianez DG, Kraft K, Heinrich V, Krawitz P, Brancati F, Klopocki E, et al. Disruptions of topological chromatin domains cause pathogenic rewiring of gene-enhancer interactions. *Cell*. 2015;161(5):1012-25.
9. Franke M, Ibrahim DM, Andrey G, Schwarzer W, Heinrich V, Schopflin R, et al. Formation of new chromatin domains determines pathogenicity of genomic duplications. *Nature*. 2016;538(7624):265-9.

REVIEWERS' COMMENTS:

Reviewer #3 (Remarks to the Author):

I am broadly happy with the revisions made. I have 3 comments (well, one of the comments is a request)

COMMENT 1: The answer to my question about the reason for excluding germline mutations in the genetic analysis is not really answered in the paper, only in the review.

COMMENT 2 (REQUEST): I would really like to have the following number somewhere in the paper (Results or Methods, not Supplemental Material): how many interactions are tested in the HiC data analysis? That is not easy to figure out and it would be trivial to change the sentence

"and found 981 significantly different interactions between MCF7 and tamoxifen-resistant TAMR cells (diffHiC, FDR < 0.05, Supplementary Table 2) and 2,596 significantly differential interactions between MCF7 and fulvestrant-resistant FASR cells (diffHiC, FDR < 0.05, Supplementary Table 3) at 20kb resolution."

to something like

"and found 981 significantly different interactions between MCF7 and tamoxifen-resistant TAMR cells (diffHiC, FDR < 0.05, X tests performed, Supplementary Table 2) and 2,596 significantly differential interactions between MCF7 and fulvestrant-resistant FASR cells (diffHiC, FDR < 0.05, X tests performed, Supplementary Table 3) at 20kb resolution."

I thought about this previously, but I see I never wrote it down.

COMMENT 3: Given the number of bins in the HiC data matrix, I personally believe the top1000 most variable interactions is far too little.

Please find our point by point response to the REVIEWERS' COMMENTS:

Reviewer #3 (Remarks to the Author):

I am broadly happy with the revisions made. I have 3 comments (well, one of the comments is a request)

COMMENT 1: The answer to my question about the reason for excluding germline mutations in the genetic analysis is not really answered in the paper, only in the review.

Response: Information relating to exclusion of potential germline variants in MuTect2 analysis is provided in the paper in the Methods section (page 24): “*SNVs that have been previously identified in dbSNP v.146 were included if they fulfilled all the requirements of MuTect2 to be called somatic, including that they were not previously reported as germline (dbSNP SAO flag “germline”).*”

COMMENT 2 (REQUEST): I would really like to have the following number somewhere in the paper (Results or Methods, not Supplemental Material): how many interactions are tested in the HiC data analysis? That is not easy to figure out and it would be trivial to change the sentence

"and found 981 significantly different interactions between MCF7 and tamoxifen-resistant TAMR cells (diffHiC, FDR < 0.05, Supplementary Table 2) and 2,596 significantly differential interactions between MCF7 and fulvestrant-resistant FASR cells (diffHiC, FDR < 0.05, Supplementary Table 3) at 20kb resolution."

to something like

"and found 981 significantly different interactions between MCF7 and tamoxifen-resistant TAMR cells (diffHiC, FDR < 0.05, X tests performed, Supplementary Table 2) and 2,596 significantly differential interactions between MCF7 and fulvestrant-resistant FASR cells (diffHiC, FDR < 0.05, X tests performed, Supplementary Table 3) at 20kb resolution."

I thought about this previously, but I see I never wrote it down.

Response: As requested, we have now included the total number of interactions (after direct filtering) interrogated in the diffHiC analysis in the Methods section (page 21): “*The statistical framework of the edgeR package was used on the final InteractionSet consisting of 145,109 regions to model the biological variability and to test for significance of identified differential genomic interactions. Differential interactions (DIs) were identified at 20kb resolution with FDR cut-off of 5%.*”

COMMENT 3: Given the number of bins in the HiC data matrix, I personally believe the top1000 most variable interactions is far too little.

Response: The distance between each pair of samples in the MDS plot is the ‘leading log fold change’, defined as the root-mean-square average of the 1000 largest log₂ fold changes between that pair of samples. Therefore, a plot using a small set of top bins visualizes the most extreme differences (e.g. top 100) whereas a plot using a large set of top bins visualizes overall differences (e.g. top 1000 as used in the study). Therefore for this analysis the top1000 most variable interactions is not “far too little” but appropriate.

We trust that we have now addressed all of the remaining comments from the reviewers.